# Bi-level Score Matching for Learning Energy-based Latent Variable Models

**Fan Bao**[*]**, Chongxuan Li**[*]**, Kun Xu, Hang Su**[†]**, Jun Zhu**[†]**, Bo Zhang**
Dept. of Comp. Sci. & Tech., Institute for AI, THBI Lab, BNRist Center,
State Key Lab for Intell. Tech. & Sys., Tsinghua University, Beijing, China
bf19@mails.tsinghua.edu.cn,{chongxuanli1991, kunxu.thu}@gmail.com,
{suhangss, dcszj, dcszb}@tsinghua.edu.cn

## Abstract

Score matching (SM) [24] provides a compelling approach to learn energy-based models (EBMs) by avoiding the calculation of partition function. However, it remains largely open to learn energy-based latent variable models (EBLVMs), except some special cases. This paper presents a bi-level score matching (BiSM) method to learn EBLVMs with general structures by reformulating SM as a bi-level optimization problem. The higher level introduces a variational posterior of the latent variables and optimizes a modified SM objective, and the lower level optimizes the variational posterior to fit the true posterior. To solve BiSM efficiently, we develop a stochastic optimization algorithm with gradient unrolling. Theoretically, we analyze the consistency of BiSM and the convergence of the stochastic algorithm. Empirically, we show the promise of BiSM in Gaussian restricted Boltzmann machines and highly nonstructural EBLVMs parameterized by deep convolutional neural networks. BiSM is comparable to the widely adopted contrastive divergence and SM methods when they are applicable; and can learn complex EBLVMs with intractable posteriors to generate natural images.

## 1 Introduction

An energy-based model (EBM) [35] employs an energy function mapping a configuration of variables to a scalar to define a Gibbs distribution, whose density is proportional to the exponential negative energy. Being flexible, EBMs can naturally incorporate latent variables to fit complex data and extract features. Among them, representative models including restricted Boltzmann machines (RBMs) [21], deep belief networks (DBNs) [23] and deep Boltzmann machines (DBMs) [48] have been widely adopted [60, 55]. However, it is challenging to learn EBMs because of the presence of the partition function, which is an integral over all possible configurations, especially when latent variables present.

The most widely used training approach is the maximum likelihood estimate (MLE), or equivalently minimizing the KL divergence. Such methods often adopt Markov chain Monte Carlo (MCMC) [42] or variational inference (VI) [29] to estimate the partition function (or its gradient with respect to the model parameters). Contrastive divergence (CD) [21] and its variants [57, 63, 43, 44, 9, 10, 16] are proven effective in models with fully visible variables or tractable posteriors of latent variables (e.g., RBMs). The recent work [63] present an unbiased version of contrastive divergence. Besides, Ingraham and Marks [26] perform approximate Bayesian inference over EBMs. In deep models such as DBNs and DBMs, previous work [23, 36, 48] often adopts layer-wise training strategy. Recently, several methods [34, 37] attempt to learn general energy-based latent variable models (EBLVMs) in a black-box manner by VI. In these methods, the problem of inferring the latent variables is addressed

---

[*]Equal contribution. [†] Corresponding author.

by advances in amortized inference [31] but the variational bounds for the partition function are either of high-bias [37] or high-variance [34] on high-dimensional data.

Score matching (SM) [24] provides a promising alternative approach to learning EBMs. Compared with MLE, SM does not need to access the partition function because of its foundation on Fisher divergence minimization [28], while involves a second order derivative. Many attempts [32, 59, 51, 54, 38, 45] try to estimate the second order derivative efficiently and recent work [38] can scale up to natural images. However, it is much more challenging to incorporate latent variables in SM than in MLE because of its specific form. As far as we know, extensions of SM for EBLVMs [56, 58] make strong structural assumptions that the posterior is tractable [56] or in the exponential family [58].

Considering the complementary advantages between MLE and SM, a natural question arises:

*Can we infer the latent variables in nonstructural EBLVMs using amortized inference as in MLE and at the same time learn such models without explicitly estimating the partition function as in SM?*

In this paper, we present bi-level score matching (BiSM), which is generally applicable to existing SM objectives [59, 54, 38] to learn EBLVMs with a minimal model assumption. The key to our approach is to reformulate a given SM objective as a bi-level optimization problem. The higher level problem modifies the original SM objective by approximating the marginal model distribution with the ratio of the model distribution over a variational posterior. The lower level problem optimizes certain divergence between the variational posterior and the true one. By reformulating the divergence used in the lower level problem, BiSM only needs to access the model energy and the variational posterior in its calculation. Further, under the nonparametric assumption [15], BiSM is equivalent to the original SM objective (see Theorem 1). To solve BiSM efficiently, we propose a practical algorithm using alternative stochastic gradient descent with gradient unrolling [41] and formally characterize the gradient bias and the convergence of the algorithm (see Theorem 2 and Corollary 3).

We evaluate BiSM on two EBLVMs. The first model is the well-known Gaussian restricted Boltzmann machine (GRBM) [60, 22]. We compare BiSM with the corresponding SM methods [56, 54] and the contrastive divergence (CD) [21, 57]. On a toy 2-D dataset and the Frey face dataset, BiSM achieves comparable performance to these strong baselines. The second model is a highly nonstructural EBLVM parameterized by deep convolutional neural networks to fit natural images. The CD and SM-based baselines are not applicable because of the intractable posteriors while BiSM can perform inference and learning successfully. We show the promise of BiSM by testing the sample quality and the inference accuracy on the MNIST and CIFAR10 datasets. To the best of our knowledge, previous state-of-the-art EBLVMs [23, 36, 48] cannot generate natural images in such a purely unsupervised learning setting.

## 2 Preliminaries

In this section, we present preliminaries about the Fisher divergence [28] and existing score matching methods in energy-based models (EBMs). Formally, an EBM defines a distribution: $p(\boldsymbol{w}; \boldsymbol{\theta}) = \tilde{p}(\boldsymbol{w}; \boldsymbol{\theta})/\mathcal{Z}(\boldsymbol{\theta}) = e^{-\mathcal{E}(\boldsymbol{w}; \boldsymbol{\theta})}/\mathcal{Z}(\boldsymbol{\theta})$, where $\mathcal{E}(\boldsymbol{w}; \boldsymbol{\theta})$ is the associated energy function parameterized by learnable parameters $\boldsymbol{\theta}$, $\tilde{p}(\boldsymbol{w}; \boldsymbol{\theta})$ is the unnormalized density, and $\mathcal{Z}(\boldsymbol{\theta}) = \int e^{-\mathcal{E}(\boldsymbol{w}; \boldsymbol{\theta})} d\boldsymbol{w}$ is the partition function. Here, we assume that the variable $\boldsymbol{w}$ is fully visible and continuous.

**Fisher divergence** The Fisher divergence [28] between the empirical data distribution $p_D(\boldsymbol{w})$ and the model distribution $p(\boldsymbol{w}; \boldsymbol{\theta})$ is defined as:

$$\mathcal{D}_F(p_D(\boldsymbol{w})||p(\boldsymbol{w}; \boldsymbol{\theta})) \triangleq \frac{1}{2}\mathbb{E}_{p_D(\boldsymbol{w})}\left[||\nabla_{\boldsymbol{w}} \log p(\boldsymbol{w}; \boldsymbol{\theta}) - \nabla_{\boldsymbol{w}} \log p_D(\boldsymbol{w})||_2^2\right], \qquad (1)$$

where $\nabla_{\boldsymbol{w}} \log p(\boldsymbol{w}; \boldsymbol{\theta})$ and $\nabla_{\boldsymbol{w}} \log p_D(\boldsymbol{w})$ are the model score function and data score function [24], respectively. The model score function does not depend on the value of $\mathcal{Z}(\boldsymbol{\theta})$. Indeed, we have:

$$\nabla_{\boldsymbol{w}} \log p(\boldsymbol{w}; \boldsymbol{\theta}) = \nabla_{\boldsymbol{w}} \log \tilde{p}(\boldsymbol{w}; \boldsymbol{\theta}) - \nabla_{\boldsymbol{w}} \log \mathcal{Z}(\boldsymbol{\theta}) = \nabla_{\boldsymbol{w}} \log \tilde{p}(\boldsymbol{w}; \boldsymbol{\theta}),$$

which makes the Fisher divergence suitable for learning EBMs.

**Score matching** To get rid of the unknown $\nabla_{\boldsymbol{w}} \log p_D(\boldsymbol{w})$ in the Fisher divergence, Hyvärinen [24] proposes an equivalent form, named score matching (SM), as follows:

$$\mathcal{J}_{SM}(\boldsymbol{\theta}) \triangleq \mathbb{E}_{p_D(\boldsymbol{w})}\left[\frac{1}{2}||\nabla_{\boldsymbol{w}} \log \tilde{p}(\boldsymbol{w}; \boldsymbol{\theta})||_2^2 + \mathrm{tr}(\nabla_{\boldsymbol{w}}^2 \log \tilde{p}(\boldsymbol{w}; \boldsymbol{\theta}))\right] \equiv \mathcal{D}_F(p_D(\boldsymbol{w})||p(\boldsymbol{w}; \boldsymbol{\theta})), \quad (2)$$

where $\nabla_{\boldsymbol{w}}^2 \log \tilde{p}(\boldsymbol{w}; \boldsymbol{\theta})$ is the Hessian matrix, $\text{tr}(\cdot)$ is the trace of a given matrix and $\equiv$ means equivalence in parameter optimization. Though elegant, a straightforward application of SM is inefficient, as the computation of $\text{tr}(\nabla_{\boldsymbol{w}}^2 \log \tilde{p}(\boldsymbol{w}; \boldsymbol{\theta}))$ is time-consuming on high-dimensional data.

**Sliced score matching** To scale up SM, Song et al. [54] propose sliced score matching (SSM):

$$\mathcal{J}_{SSM}(\boldsymbol{\theta}) \triangleq \frac{1}{2} \mathbb{E}_{p_D(\boldsymbol{w})} \left[ ||\nabla_{\boldsymbol{w}} \log \tilde{p}(\boldsymbol{w}; \boldsymbol{\theta})||_2^2 \right] + \mathbb{E}_{p_D(\boldsymbol{w})} \mathbb{E}_{p(\boldsymbol{u})} \left[ \boldsymbol{u}^\top \nabla_{\boldsymbol{w}}^2 \log \tilde{p}(\boldsymbol{w}; \boldsymbol{\theta}) \boldsymbol{u} \right], \quad (3)$$

where $\boldsymbol{u}$ is a random variable that is independent of $\boldsymbol{w}$ and $p(\boldsymbol{u})$ satisfies certain mild conditions [54] to ensure that SSM is consistent with SM. Instead of calculating the trace of the Hessian matrix in SM, SSM computes the product of the Hessian matrix and a vector, which can be efficiently implemented by taking two normal back-propagation processes.

**Denoising score matching** Denoising score matching (DSM) [59] is another fast variant of SM:

$$\mathcal{J}_{DSM}(\boldsymbol{\theta}) \triangleq \mathbb{E}_{p_D(\boldsymbol{w}) p_\sigma(\tilde{\boldsymbol{w}}|\boldsymbol{w})} ||\nabla_{\tilde{\boldsymbol{w}}} \log \tilde{p}(\tilde{\boldsymbol{w}}; \boldsymbol{\theta}) - \nabla_{\tilde{\boldsymbol{w}}} \log p_\sigma(\tilde{\boldsymbol{w}}|\boldsymbol{w})||_2^2 \equiv \mathcal{D}_F(p_\sigma(\tilde{\boldsymbol{w}})||p(\tilde{\boldsymbol{w}}; \boldsymbol{\theta})), \quad (4)$$

where $\tilde{\boldsymbol{w}}$ is the data perturbed by a noise disitribution $p_\sigma(\tilde{\boldsymbol{w}}|\boldsymbol{w})$ with a hyperparameter $\sigma$ and $p_\sigma(\tilde{\boldsymbol{w}}) = \int p_D(\boldsymbol{w}) p_\sigma(\tilde{\boldsymbol{w}}|\boldsymbol{w}) d\boldsymbol{w}$. A commonly chosen perturbation distribution is the Gaussian one that $p_\sigma(\tilde{\boldsymbol{w}}|\boldsymbol{w}) = \mathcal{N}(\tilde{\boldsymbol{w}}|\boldsymbol{w}, \sigma^2 I)$. DSM optimizes $\mathcal{D}_F(p_\sigma(\tilde{\boldsymbol{w}})||p(\tilde{\boldsymbol{w}}; \boldsymbol{\theta}))$ and is slightly inconsistent.

**Multiscale denoising score matching** Recently, Li et al. [38] propose multiscale denoising score matching (MDSM) to leverage different levels of noise to train EBMs on high-dimensional data as:

$$\mathcal{J}_{MDSM}(\boldsymbol{\theta}) \triangleq \mathbb{E}_{p_D(\boldsymbol{w}) p(\sigma) p_\sigma(\tilde{\boldsymbol{w}}|\boldsymbol{w})} ||\nabla_{\tilde{\boldsymbol{w}}} \log \tilde{p}(\tilde{\boldsymbol{w}}; \boldsymbol{\theta}) - \nabla_{\tilde{\boldsymbol{w}}} \log p_{\sigma_0}(\tilde{\boldsymbol{w}}|\boldsymbol{w})||_2^2, \quad (5)$$

where $p(\sigma)$ is a prior distribution over the noise levels and $\sigma_0$ is a fixed noise level.

# 3 Method

In this paper, we aim to extend the above SM methods to learn general energy-based latent variable models (EBLVMs). In contrast to previous work [56, 58], our method only accesses the energy function without any structural assumption of the model. Formally, an EBLVM defines a probability distribution over a set of continuous visible variables $\boldsymbol{v}$ [2] and a set of latent variables $\boldsymbol{h}$ as follows:

$$p(\boldsymbol{v}, \boldsymbol{h}; \boldsymbol{\theta}) = \tilde{p}(\boldsymbol{v}, \boldsymbol{h}; \boldsymbol{\theta}) / \mathcal{Z}(\boldsymbol{\theta}) = e^{-\mathcal{E}(\boldsymbol{v}, \boldsymbol{h}; \boldsymbol{\theta})} / \mathcal{Z}(\boldsymbol{\theta}), \quad (6)$$

where $\mathcal{E}(\boldsymbol{v}, \boldsymbol{h}; \boldsymbol{\theta})$ is the associated energy function with learnable parameters $\boldsymbol{\theta}$, $\tilde{p}(\boldsymbol{v}, \boldsymbol{h}; \boldsymbol{\theta})$ is the unnormalized density, and $\mathcal{Z}(\boldsymbol{\theta}) = \int e^{-\mathcal{E}(\boldsymbol{v}, \boldsymbol{h}; \boldsymbol{\theta})} d\boldsymbol{v} d\boldsymbol{h}$ is the partition function. In general, the marginal distribution $p(\boldsymbol{v}; \boldsymbol{\theta})$ and the posterior distribution $p(\boldsymbol{h}|\boldsymbol{v}; \boldsymbol{\theta})$ are intractable.

We would like to minimize $\mathcal{D}_F(q(\boldsymbol{v})||p(\boldsymbol{v}; \boldsymbol{\theta}))$, namely, the Fisher divergence between the marginal model distribution $p(\boldsymbol{v}; \boldsymbol{\theta})$ and $q(\boldsymbol{v})$, which can be the empirical data distribution $p_D(\boldsymbol{v})$ as in SM [24] or the perturbed one $p_\sigma(\tilde{\boldsymbol{v}}) = \int p_D(\boldsymbol{v}) p_\sigma(\tilde{\boldsymbol{v}}|\boldsymbol{v}) d\boldsymbol{v}$ in DSM [59]. Equivalently, we can optimize a certain SM objective in Eqn. (2-5), which is generally expressed in the following form:

$$\mathcal{J}(\boldsymbol{\theta}) = \mathbb{E}_{q(\boldsymbol{v}, \boldsymbol{\epsilon})} \mathcal{F}(\nabla_{\boldsymbol{v}} \log p(\boldsymbol{v}; \boldsymbol{\theta}), \boldsymbol{\epsilon}, \boldsymbol{v}), \quad (7)$$

where $\mathcal{F}$ is a functional that depends on which SM objective we choose, $\boldsymbol{\epsilon}$ is introduced to represent additional random noise used in SSM [54] or DSM [59], and $q(\boldsymbol{v}, \boldsymbol{\epsilon})$ denotes the joint distribution of $\boldsymbol{v}$ and $\boldsymbol{\epsilon}$. The same challenge for all SM objectives is that the marginal score function $\nabla_{\boldsymbol{v}} \log p(\boldsymbol{v}; \boldsymbol{\theta})$ is intractable and we propose bi-level score matching (BiSM) to solve the problem in this paper.

## 3.1 Bi-level Score Matching

First, we notice that the marginal score function can be rewritten as $\nabla_{\boldsymbol{v}} \log p(\boldsymbol{v}; \boldsymbol{\theta}) = \nabla_{\boldsymbol{v}} \log \frac{\tilde{p}(\boldsymbol{v}, \boldsymbol{h}; \boldsymbol{\theta})}{p(\boldsymbol{h}|\boldsymbol{v}; \boldsymbol{\theta})} - \nabla_{\boldsymbol{v}} \log \mathcal{Z}(\boldsymbol{\theta}) = \nabla_{\boldsymbol{v}} \log \frac{\tilde{p}(\boldsymbol{v}, \boldsymbol{h}; \boldsymbol{\theta})}{p(\boldsymbol{h}|\boldsymbol{v}; \boldsymbol{\theta})}$. We introduce a variational posterior distribution $q(\boldsymbol{h}|\boldsymbol{v}; \boldsymbol{\phi})$ to approximate the true posterior $p(\boldsymbol{h}|\boldsymbol{v}; \boldsymbol{\theta})$, and obtain an approximation of the marginal score function using $\nabla_{\boldsymbol{v}} \log \frac{\tilde{p}(\boldsymbol{v}, \boldsymbol{h}; \boldsymbol{\theta})}{q(\boldsymbol{h}|\boldsymbol{v}; \boldsymbol{\theta})}$. We reformulate the general SM objective in Eqn. (7) as the following bi-level optimization problem:

$$\boldsymbol{\theta}^* = \underset{\boldsymbol{\theta} \in \Theta}{\arg\min} \, \mathcal{J}_{Bi}(\boldsymbol{\theta}, \boldsymbol{\phi}^*(\boldsymbol{\theta})), \quad \mathcal{J}_{Bi}(\boldsymbol{\theta}, \boldsymbol{\phi}) = \mathbb{E}_{q(\boldsymbol{v}, \boldsymbol{\epsilon})} \mathbb{E}_{q(\boldsymbol{h}|\boldsymbol{v}; \boldsymbol{\phi})} \mathcal{F} \left( \nabla_{\boldsymbol{v}} \log \frac{\tilde{p}(\boldsymbol{v}, \boldsymbol{h}; \boldsymbol{\theta})}{q(\boldsymbol{h}|\boldsymbol{v}; \boldsymbol{\phi})}, \boldsymbol{\epsilon}, \boldsymbol{v} \right), \quad (8)$$

where $\Theta$ is the hypothesis space of the model and $\phi^*(\boldsymbol{\theta})$ is defined as follows:

$$\phi^*(\boldsymbol{\theta}) = \underset{\phi \in \Phi}{\arg\min}\, \mathcal{G}(\boldsymbol{\theta}, \phi), \text{ with } \mathcal{G}(\boldsymbol{\theta}, \phi) = \mathbb{E}_{q(\boldsymbol{v}, \boldsymbol{\epsilon})} \mathcal{D}\left(q(\boldsymbol{h}|\boldsymbol{v}; \phi) || p(\boldsymbol{h}|\boldsymbol{v}; \boldsymbol{\theta})\right), \qquad (9)$$

where $\Phi$ is the hypothesis space of the variational posterior and $\mathcal{D}$ is a certain divergence to be specified later. We denote $\phi^*$ as a function of $\boldsymbol{\theta}$ to explicitly present the dependency. We emphasize that the bi-level formulation is necessary because if we treat $\phi^*$ as a constant with respect to $\boldsymbol{\theta}$, then $\nabla_{\boldsymbol{\theta}} \mathcal{J}(\boldsymbol{\theta}) \neq \nabla_{\boldsymbol{\theta}} \mathcal{J}_{Bi}(\boldsymbol{\theta}, \phi^*)$ in general. In contrast, the equivalence of the bi-level formulation in Eqn. (8-9) and SM under the nonparametric assumption [15] is characterized in Theorem 1.

**Theorem 1.** *(Equivalence of BiSM, proof in Appendix A.1) Assuming that $\forall \boldsymbol{\theta} \in \Theta$, $\exists \phi \in \Phi$ such that $\mathcal{D}(q(\boldsymbol{h}|\boldsymbol{v}; \phi) || p(\boldsymbol{h}|\boldsymbol{v}; \boldsymbol{\theta})) = 0, \forall \boldsymbol{v} \in \text{supp}(q)$, we have $\nabla_{\boldsymbol{\theta}} \mathcal{J}(\boldsymbol{\theta}) = \nabla_{\boldsymbol{\theta}} \mathcal{J}_{Bi}(\boldsymbol{\theta}, \phi^*(\boldsymbol{\theta}))$.*

We notice that such assumptions have been made in the typical variational inference [29] to obtain a tight estimate of the log-likelihood and the generative adversarial networks [15] to guarantee validness. Further, in a practical case with a less powerful $q(\boldsymbol{h}|\boldsymbol{v}; \phi)$, we bound the bias of BiSM by the approximation error of $q(\boldsymbol{h}|\boldsymbol{v}; \phi)$ in Appendix A.1 to provide a complementary analysis.

To learn general EBLVMs with intractable posteriors, the lower level optimization problem in Eqn. (9) can only access the unnormalized model distribution $\tilde{p}(\boldsymbol{v}, \boldsymbol{h}; \boldsymbol{\theta})$ and the variational posterior $q(\boldsymbol{h}|\boldsymbol{v}; \phi)$ in calculation, as in the high-level problem in Eqn. (8). We first consider the widely adopted KL divergence for variational inference and obtain an equivalent form regarding to optimizing $\phi$:

$$\mathcal{D}_{KL}\left(q(\boldsymbol{h}|\boldsymbol{v}; \phi) || p(\boldsymbol{h}|\boldsymbol{v}; \boldsymbol{\theta})\right) \equiv \mathbb{E}_{q(\boldsymbol{h}|\boldsymbol{v}; \phi)} \log \frac{q(\boldsymbol{h}|\boldsymbol{v}; \phi)}{\tilde{p}(\boldsymbol{v}, \boldsymbol{h}; \boldsymbol{\theta})}, \qquad (10)$$

from which an unknown constant is subtracted. Therefore, Eqn. (10) is sufficient for training $\phi$ but not suitable for evaluating the inference accuracy. In contrast, the Fisher divergence, which is an alternative approach for variational inference [62], can be directly calculated by:

$$\mathcal{D}_F\left(q(\boldsymbol{h}|\boldsymbol{v}; \phi) || p(\boldsymbol{h}|\boldsymbol{v}; \boldsymbol{\theta})\right) = \frac{1}{2} \mathbb{E}_{q(\boldsymbol{h}|\boldsymbol{v}; \phi)} \left[ ||\nabla_{\boldsymbol{h}} \log q(\boldsymbol{h}|\boldsymbol{v}; \phi) - \nabla_{\boldsymbol{h}} \log \tilde{p}(\boldsymbol{v}, \boldsymbol{h}; \boldsymbol{\theta})||_2^2 \right]. \qquad (11)$$

For the detailed derivation, please see Appendix A.2. Compared with the KL divergence in Eqn. (10), the Fisher divergence in Eqn. (11) can be used for both training and evaluation but cannot deal with discrete $\boldsymbol{h}$ in which case $\nabla_{\boldsymbol{h}}$ is not well defined. In our experiments, we apply them according to the specific scenario. In principle, any other divergence that does not necessarily access $p(\boldsymbol{v}; \boldsymbol{\theta})$ or $p(\boldsymbol{h}|\boldsymbol{v}; \boldsymbol{\theta})$ can be used here and we leave a systematical study of the divergence for the future work.

### 3.2 Stochastic Optimization for BiSM

Our goal is to learn general EBLVMs whose energy can be parameterized by highly nonlinear and nonstructural functions, e.g. deep neural networks (DNNs). Such models are rarely studied before due to the challenges in training and inference. In Sec. 5.2, we propose an instance motivated by the fully visible EBMs [9, 38] to fit natural images and validate the effectiveness of BiSM. To fit the intractable posterior, we also employ a variational posterior parameterized by DNNs. In this context, it is impractical to exactly solve the BiSM problem in Eqn. (8-9) on full batch. Therefore, we develop a practical algorithm for BiSM by updating $\phi$ and $\boldsymbol{\theta}$ alternatively using stochastic gradient descent. Formally, assuming a minibatch of data is given, let $\hat{\mathcal{J}}(\boldsymbol{\theta})$, $\hat{\mathcal{J}}_{Bi}(\boldsymbol{\theta}, \phi)$ and $\hat{\mathcal{G}}(\boldsymbol{\theta}, \phi)$ denote the corresponding functions evaluated on the minibatch. Motivated by our analysis in Theorem 2 presented later, we first update $\phi$ for $K$ times on the same minibatch of data by:

$$\phi \leftarrow \phi - \alpha \frac{\partial \hat{\mathcal{G}}(\boldsymbol{\theta}, \phi)}{\partial \phi}, \qquad (12)$$

where $\alpha$ is a prefixed learning rate scheme. We denote the resulting parameters of the variational posterior as $\phi^0$, which approximates $\hat{\phi}^*(\boldsymbol{\theta}) = \arg\min_{\phi \in \Phi} \hat{\mathcal{G}}(\boldsymbol{\theta}, \phi)$. To update $\boldsymbol{\theta}$, the central challenge is to approximate the stochastic gradient $\frac{\partial \hat{\mathcal{J}}_{Bi}(\boldsymbol{\theta}, \hat{\phi}^*(\boldsymbol{\theta}))}{\partial \boldsymbol{\theta}}$, which is addressed by the gradient unrolling technique [41]. Specifically, we start from $\phi^0$ and calculate $\hat{\phi}^N(\boldsymbol{\theta})$ recursively by:

$$\hat{\phi}^1(\boldsymbol{\theta}) = \phi^0 - \alpha \frac{\partial \hat{\mathcal{G}}(\boldsymbol{\theta}, \phi)}{\partial \phi}\Big|_{\phi = \phi^0}, \text{ and } \hat{\phi}^n(\boldsymbol{\theta}) = \hat{\phi}^{n-1}(\boldsymbol{\theta}) - \alpha \frac{\partial \hat{\mathcal{G}}(\boldsymbol{\theta}, \phi)}{\partial \phi}\Big|_{\phi = \hat{\phi}^{n-1}(\boldsymbol{\theta})}, \qquad (13)$$

for $n = 2, ..., N$, where we treat $\phi^0$ as a constant with respect to $\boldsymbol{\theta}$, and make the dependence of $\hat{\phi}^n(\boldsymbol{\theta})$ on $\phi^0$ implicit for simplicity. Note that Eqn. (13) is not used to update the variational parameters but to approximate $\frac{\partial \hat{\mathcal{J}}_{Bi}(\boldsymbol{\theta}, \hat{\phi}^*(\boldsymbol{\theta}))}{\partial \boldsymbol{\theta}}$ by the gradient of a surrogate loss $\hat{\mathcal{J}}_{Bi}(\boldsymbol{\theta}, \hat{\phi}^N(\boldsymbol{\theta}))$:

$$\frac{\partial \hat{\mathcal{J}}_{Bi}(\boldsymbol{\theta}, \hat{\phi}^N(\boldsymbol{\theta}))}{\partial \boldsymbol{\theta}} = \frac{\partial \hat{\mathcal{J}}_{Bi}(\boldsymbol{\theta}, \phi)}{\partial \boldsymbol{\theta}}|_{\phi=\hat{\phi}^N(\boldsymbol{\theta})} + \frac{\partial \hat{\mathcal{J}}_{Bi}(\boldsymbol{\theta}, \phi)}{\partial \phi}|_{\phi=\hat{\phi}^N(\boldsymbol{\theta})} \frac{\partial \hat{\phi}^N(\boldsymbol{\theta})}{\partial \boldsymbol{\theta}}. \tag{14}$$

Given the above stochastic gradient, we update the parameters in the model distribution by:

$$\boldsymbol{\theta} \leftarrow \boldsymbol{\theta} - \beta \frac{\partial \hat{\mathcal{J}}_{Bi}(\boldsymbol{\theta}, \hat{\phi}^N(\boldsymbol{\theta}))}{\partial \boldsymbol{\theta}}, \tag{15}$$

where $\beta$ is a prefixed learning rate scheme. The whole training procedure is summarized in Algorithm 1. The gradient unrolling technique has been proposed to learn implicit directed generative models [41, 9]. In comparison, we learn EBMs and construct a different bi-level optimization problem based on the Fisher divergence. Further, below we formally analyze the approximation error of the stochastic gradient, which has not been explored in the related work [41, 9].

**Theorem 2.** *(Approximate error of the stochastic gradient, proof in Appendix A.4) Supposing that:*

1. *Both $\Theta$ and $\Phi$ are compact and convex,*

2. *$\hat{\mathcal{J}}_{Bi}(\boldsymbol{\theta}, \phi) \in C^2(\Omega)$, $\hat{\mathcal{G}}(\boldsymbol{\theta}, \phi) \in C^3(\Omega)$, where $\Omega$ is an open set including $\Theta \times \Phi$ (i.e. $\hat{\mathcal{J}}_{Bi}$ and $\hat{\mathcal{G}}$ are second and third order continuously differentiable on $\Omega$ respectively),*

3. *$\hat{\mathcal{G}}(\boldsymbol{\theta}, \phi)$ is strongly convex on $\Phi$ for all $\boldsymbol{\theta} \in \Theta$,*

4. *$\forall N \geq 0, \forall \boldsymbol{\theta} \in \Theta, \forall \phi^0 \in \Phi, \hat{\phi}^N(\boldsymbol{\theta}, \phi^0) \in \Phi$ and $\hat{\phi}^*(\boldsymbol{\theta}) \in \Phi$,*

*then when $\alpha$ is small enough, there exists $A, B, C > 0$ and $\kappa \in (0, 1)$ independent of $\boldsymbol{\theta}$ and $\phi^0$, s.t.,*

$$||\frac{\partial \hat{\mathcal{J}}_{Bi}(\boldsymbol{\theta}, \hat{\phi}^N(\boldsymbol{\theta}, \phi^0))}{\partial \boldsymbol{\theta}} - \frac{\partial \hat{\mathcal{J}}_{Bi}(\boldsymbol{\theta}, \hat{\phi}^*(\boldsymbol{\theta}))}{\partial \boldsymbol{\theta}}|| \leq (A + BN)\kappa^N ||\phi^0 - \hat{\phi}^*(\boldsymbol{\theta})|| + C\kappa^N,$$

*for all $\boldsymbol{\theta} \in \Theta$, $\phi^0 \in \Phi$ and $N \geq 0$.*

Theorem 2 implies that the approximation error converges to zero in a linear rate in terms of $N$ when $\hat{\mathcal{G}}$ is strongly convex, which is a commonly used assumption to obtain such results [3]. Although the assumption does not always hold in our experiments, Theorem 2 still provides insights into our implementation. As mentioned before, we update $\phi$ for $K$ times on the same minibatch of data to reduce $||\phi^0 - \hat{\phi}^*||$. In Fig. 1, we numerically validate Theorem 2. The gradient bias decays (approximately) exponentially w.r.t. $N$, which is consistent with Theorem 2. As for $||\phi^0 - \hat{\phi}^*||$, we find it decreases from 1.38 to 0.87 as $K$ increases from 0 to 20. It leads to smaller bias, which agrees with Fig. 1 and Theorem 2. Besides, we notice that the unrolling technique is not exclusive for advances in non-convex optimization [3] and we leave the general analysis for the future work. Due to the controllable approximation error, the convergence of BiSM can be formally characterized as follows.

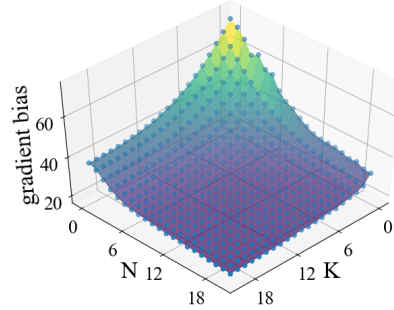

Figure 1: The gradient bias (the left hand side of Theorem 2) w.r.t. $N$ and $K$ in GRBM on Frey face.

**Corollary 3.** *(BiSM finds $\delta$-stationary points, proof in Appendix A.5) For any accuracy level $\delta > 0$, assuming Theorem 2 holds, using a sufficiently large $N$, i.e. asymptotically $\mathcal{O}(\log \frac{1}{\delta})$, and a proper learning rate scheme $\beta$ [3], Algorithm 1 converges to a $\delta$-stationary point of BiSM in Eqn. (8-9), and further a $\delta$-stationary point of SM in Eqn. (7) if Theorem 1 also holds.*

## 4  Related Work

Apart from the MLE and SM-based methods mentioned before, we present other related work on learning and inference in energy-based latent variable models (EBLVMs) and bi-level optimization.

Previous state-of-the-art deep EBLVMs include deep belief networks (DBNs) [23], convolutional deep belief networks (CDBNs) [36] and deep Boltzmann machines (DBMs) [48]. In comparison, first, their unsupervised pretraining algorithms explicitly leverage the layer-wise structures in their models. In contrast, BiSM is an end-to-end approach which does not require any specific structural assumption. A direct comparison of BiSM and the layer-wise algorithms in DBNs and DBMs is nontrivial because the model likelihoods are discrete and then the Fisher divergence is not well-defined. We mention that extensions of SM [25, 39, 52] can deal with discrete data and BiSM can be applied to such methods as well but we leave it for the

---

**Algorithm 1** Bi-level score matching by alternative stochastic gradient descent

---

1: **Input:** Constants $K$ and $N$, learning rate schemes $\alpha$ and $\beta$, randomly initialized $\boldsymbol{\theta}$ and $\boldsymbol{\phi}$
2: **repeat**
3:     Sample a minibatch of data
4:     **for** $i = 1, ..., K$ **do**
5:         Update $\boldsymbol{\phi}$ according to Eqn. (12)
6:     **end for**
7:     $\boldsymbol{\phi}^0 \leftarrow \boldsymbol{\phi}$
8:     **for** $n = 1, ..., N$ **do**
9:         Compute $\hat{\boldsymbol{\phi}}^N(\boldsymbol{\theta}, \boldsymbol{\phi}^0)$ according to Eqn. (13)
10:     **end for**
11:     Approximate the stochastic gradient according to Eqn. (14) and update $\boldsymbol{\theta}$ according to Eqn. (15)
12: **until** Convergence or reaching certain threshold

---

future work. Second, these methods either focus on supervised learning [36] or model relatively simple data [23, 48, 49]. Our experiments show a deep EBLVM trained by BiSM can successfully generate natural images in a purely unsupervised learning setting, which has not been explored before. Besides, it is possible to generalize BiSM to (semi-)supervised learning like the extensions [33, 50, 6] of directed deep generative models [31, 15].

Noise contrastive estimation (NCE) [17] is another criterion to learn unnormalized models. It discriminates samples from the model distribution and a prefixed noise distribution. The main bottleneck of NCE on high-dimensional data is how to define a proper noise distribution manually. Variational noise-contrastive estimation [47] applies variational inference to the NCE objective and therefore can be used for posterior inference of latent variables. However, the difficulty of choosing the noise distribution remains and only small experiments are reported in VNCE [47].

The bi-level optimization problem arises in different tasks, such as learning implicit directed generative models [15, 41, 9], meta learning [12, 46, 13], and many others [40, 8, 1, 2]. Franceschi et al. [13] provide some other properties of gradient unrolling [41], from which a result similar to Corollary 3 is derived. As an alternative approach to gradient unrolling [41], implicit gradient formulation is adopted in [8, 1, 46, 2]. In comparison, implicit gradient can be accurate in a non-asymptotic setting but needs to solve a quadratic problem involving the inverse of the Hessian matrix w.r.t. the parameters in the lower level problem. Our preliminary experiments suggest that it is impractical to use the implicit gradient in BiSM if the variational posterior is a deep convolutional neural network.

## 5 Experiment

We evaluate BiSM on two EBLVMs in our experiments. The first one is the well-known Gaussian restricted Boltzmann machine (GRBM) [60, 22], which is a good benchmark to compare with existing methods [22, 24]. The second one is a deep EBLVM introduced in this paper to model natural image data. We would like to validate two arguments: (1) BiSM is comparable to strong baselines, including the contrastive divergence (CD)-based methods [21, 57] and SM-based methods [56, 54], when they are applicable; and (2) BiSM can learn general EBLVMs with neural network energy functions and generate natural images like CIFAR10, which have not been explored in previous work to our knowledge. We explicitly denote our method as *BiDSM*, *BiSSM* and *BiMDSM*, respectively, based on which SM objective [59, 54, 38] is used. All of them are collectively called *BiSM* for simplicity.

### 5.1 GRBM

The energy function of a GRBM is $\mathcal{E}(\boldsymbol{v}, \boldsymbol{h}; \boldsymbol{\theta}) = \frac{1}{2\sigma^2}||\boldsymbol{v} - b||^2 - c^\top \boldsymbol{h} - \frac{1}{\sigma}\boldsymbol{v}^\top W \boldsymbol{h}$, where the learnable parameters are $\boldsymbol{\theta} = (\sigma, W, b, c)$. To deal with discrete $\boldsymbol{h}$, we use the KL divergence in Eqn. (10) and the Gumbel-Softmax trick [27] in the lower-level problem. Because $p(\boldsymbol{h}|\boldsymbol{v}; \boldsymbol{\theta})$ is tractable, DSM [11] and SSM [54] can learn a GRBM as the way described in [56], which serve as gold standard baselines of our method. Besides, we also compare BiSM against the popular CD-based methods [21, 57] and the NCE-based methods [17, 47].

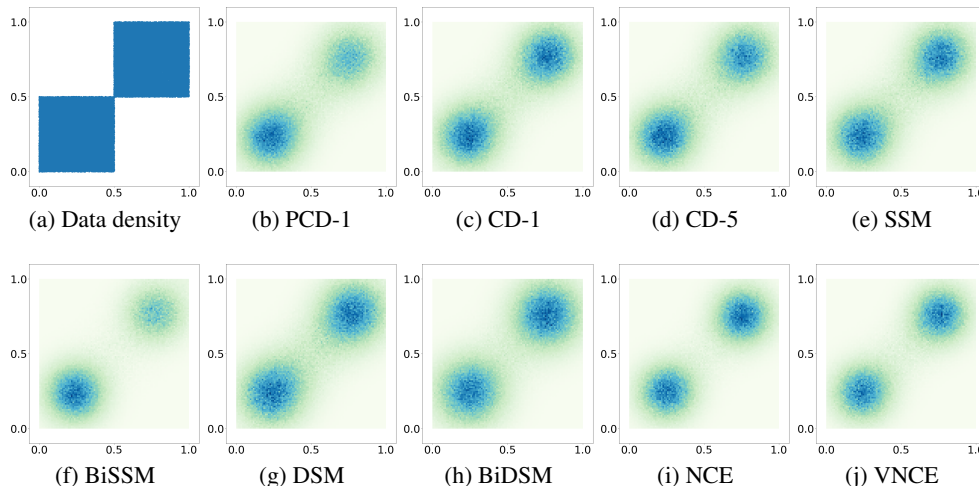

Figure 2: A small GRBM trained by different methods to fit the checkerboard dataset. Our BiSSM-5 and BiDSM-5 achieve comparable performance to the SM, CD and NCE baselines.

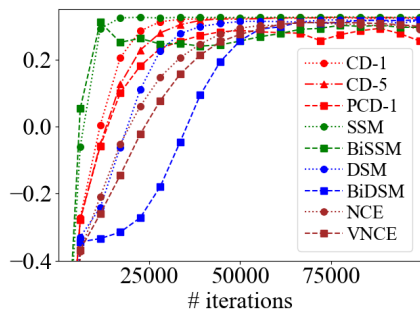

Figure 3: Test log-likelihood on the checkerboard dataset (averaged over 10 runs).

Table 1: Test log-likelihood (LL) and test Fisher divergence (Fisher) (subtracted by the same unknown constant only relevant to the data) according to the best validation performance on the Frey face dataset.

| Method | LL ↑ | Fisher ↓ |
|---|---|---|
| DSM | 129.23 | -5885.09 |
| BiDSM ($N$=0) | 107.59 | -5474.52 |
| BiDSM ($N$=1) | 110.65 | -5516.07 |
| BiDSM ($N$=5) | 124.00 | -5780.18 |
| BiDSM ($N$=10) | 125.72 | -5800.17 |

We briefly summarize the default experimental settings here, see Appendix B.1 and the source code[3] for more details. We evaluate BiSM on the checkerboard dataset and the Frey face dataset. The checkerboard dataset consists of 2-D points and the density is shown in Fig. 2 (a). We generate 60,000 points for training and 10,000 points for testing. The dimension of $h$ is 4. The Frey face dataset consists of gray-scaled face images of size $20 \times 28$. We split 1,400 images for training, 300 images for validation and 265 images for testing. The dimension of $h$ is 400. On both datasets, $q(h|v; \phi)$ is a Bernoulli distribution parameterized by a fully connected layer with the sigmoid activation and we use $K = 5$ and $N = 5$ in BiSM. We use the Adam [30] optimizer for all methods. The learning rate is $10^{-3}$ on the checkerboard dataset and $2 \times 10^{-4}$ on the Frey face dataset.

In Fig. 2, we plot the density of the same GRBM trained by different methods on the checkerboard dataset. Our BiSSM and BiDSM and the corresponding SM baselines [54, 59] are comparable after convergence as shown in Fig. 2 (e-h), demonstrating that BiSM can extend SM to deal with EBLVMs in a black-box manner without hurting performance. The results of CD-based methods in Fig. 2 (b-d) and NCE-based methods in Fig. 2 (i-j) are similar to BiSM. Besides, we calculate the test log-likelihood by brute force in Fig. 3 for a quantitative comparison. The results after convergence agree with the results in Fig. 2. The convergence speed of BiSSM is as fast as SSM while that of BiDSM is slightly slower than DSM.

We provide the results on the Frey face dataset in Tab. 1. We consider the log-likelihood as well as the score matching loss, which is the Fisher divergence subtracted by an unknown constant that is only relevant to the data distribution. In this case, SM [54] needs additive noise on the data [53] and is equivalent to DSM. CD-based methods require the model and data distribution to be properly modified [5, 4, 37], resulting in incomparable (unnormalized) Fisher divergence. Therefore, we focus

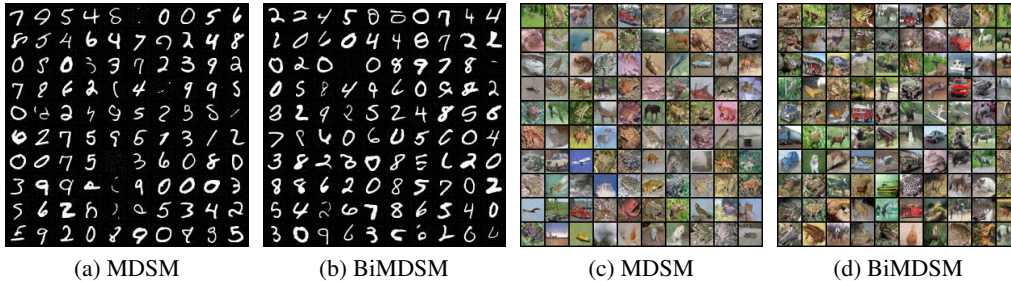

| (a) MDSM | (b) BiMDSM | (c) MDSM | (d) BiMDSM |

Figure 4: Samples from EBMs trained by MDSM and comparable EBLVMs trained by BiMDSM on the MNIST and CIFAR10 datasets. The samples from both models are of similar visual quality.

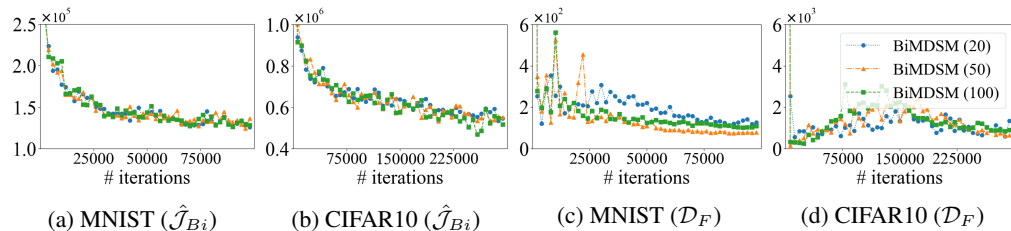

| (a) MNIST ($\hat{\mathcal{J}}_{Bi}$) | (b) CIFAR10 ($\hat{\mathcal{J}}_{Bi}$) | (c) MNIST ($\mathcal{D}_F$) | (d) CIFAR10 ($\mathcal{D}_F$) |

Figure 5: Learning curves of BiSM on MNIST and CIFAR10. Fig. (a-b) show the learning curve of $\hat{\mathcal{J}}_{Bi}(\boldsymbol{\theta}, \hat{\boldsymbol{\phi}}^N(\boldsymbol{\theta}))$. Fig. (c-d) show the learning curve of $\mathcal{D}_F(q(\boldsymbol{h}|\boldsymbol{v})||p(\boldsymbol{h}|\boldsymbol{v}))$.

on the comparison with DSM here. According to Tab. 1, as $N$ increases, BiDSM gets better due to the smaller approximation error of the stochastic gradient and BiDSM is comparable to DSM with $N \geq 5$. See Appendix C.1 for the samples, additional sensitivity analysis and time complexity comparison.

## 5.2 Deep EBLVM

We introduce a deep EBLVM with energy function $\mathcal{E}(\boldsymbol{v}, \boldsymbol{h}; \boldsymbol{\theta}) = g_3(g_2(g_1(\boldsymbol{v}; \boldsymbol{\theta}_1), \boldsymbol{h}); \boldsymbol{\theta}_2)$, where $\boldsymbol{\theta} = (\boldsymbol{\theta}_1, \boldsymbol{\theta}_2)$. $g_1(\cdot)$ is a neural network that outputs a feature sharing the same dimension with $\boldsymbol{h}$ and the architecture is adopted from the deep EBM proposed in MDSM [38]. $g_2(\cdot, \cdot)$ is an additive coupling layer [7] to make the features and the latent variables strongly coupled. $g_3(\cdot)$ is a small neural network that outputs a scalar. Because the posterior is intractable in the deep EBLVM, the baselines used in Sec.5.1 are not applicable. Further, to our knowledge, such models have not been explored in related work on learning deep EBLVMs [49, 34, 37] due to their limitations discussed before. Therefore, we use a fully visible EBM trained by MDSM [38] as our baseline. We emphasize that our goal is not to achieve state-of-the-art results on the related tasks but to validate that BiSM can learn complex EBLVMs to generate natural images and perform posterior inference accurately.

We evaluate both methods on the MNIST and CIFAR10 datasets. The MNIST dataset consists of gray-scaled hand-written digits of size $28 \times 28$ and the CIFAR10 dataset consists of color natural images of size $32 \times 32$. Following MDSM [38], we split 60,000 samples for training and 10,000 samples for testing on MNIST, and split 50,000 samples for training and 10,000 samples for testing on CIFAR10; we use a 12-layer ResNet [19] and a 18-layer ResNet in $g_1(\cdot)$ on the MNIST and CIFAR10 datasets respectively. On both datasets, we use a fully connected layer in $g_3(\cdot)$. Overall, our EBLVMs have comparable parameters to the models of the baseline [38]. On both datasets, the dimension of $\boldsymbol{h}$ is 50 by default, and $q(\boldsymbol{h}|\boldsymbol{v}; \boldsymbol{\phi})$ is a Gaussian distribution parameterized by a 3-layer convolutional neural network (CNN). We use $K = 5$ and $N = 0$ for time and memory efficiency and the Adam [30] optimizer with learning rates $10^{-4}$ and $5 \times 10^{-5}$ for training on MNIST and CIFAR10 respectively following [38]. See further details (e.g., the sampling procedure) in Appendix B.2.

In Fig.4, we present the samples on the MNIST and CIFAR10 datasets. It can be seen that our method produces samples of similar visual quality to the baseline's, indicating that the deep EBLVM learned by BiMDSM captures the marginal data distribution. Tab. 2 shows the FID [20] scores of existing EBMs. Following a similar protocol adopted in MDSM [38], the third column shows the FID with early stopping (ES) according to the results on 1,000 samples. We also implement MDSM in our

code for a fair comparison. The reproduced MDSM is slightly better than the original paper [38] and serves as a stronger baseline. Our result outperforms MDSM in a fair comparison, indicating that introducing latent variables can improve the sample quality. We mention that Flow-CE [14] and VAE-EBLVM [18] generate samples from a flow and a VAE respectively instead of EBMs are less comparable. We include them because such results have not been reported in previous state-of-the-art EBLVMs [23, 49, 36]. We hope our results can serve as a benchmark for the future work on EBLVMs.

As for posterior inference (the lower level problem), Fig. 5 (c-d) shows the learning curves of the Fisher divergence (see Eqn. (11)) between the variational posterior $q(\boldsymbol{h}|\boldsymbol{v})$ and the model posterior $p(\boldsymbol{h}|\boldsymbol{v})$ with different dimensions of $\boldsymbol{h}$ on the MNIST and CIFAR10 datasets. In all settings, the divergence decreases from a magnitude of $10^5$ or $10^4$ to a magnitude of $10^2$ or $10^1$, suggesting that $q(\boldsymbol{h}|\boldsymbol{v})$ learned by BiSM is an accurate approximation of $p(\boldsymbol{h}|\boldsymbol{v})$. On CIFAR10, the divergence sometimes increases perhaps because $p(\boldsymbol{h}|\boldsymbol{v})$ gets complex during training. Nevertheless, the divergence after

Table 2: FID on CIFAR10. $^\dagger$ means averaged by 5 runs. Methods with $^\ddagger$ use comparable networks.

| Method | FID $\downarrow$ | FID-ES $\downarrow$ |
|---|---|---|
| Flow-CE [14] | 37.30 | - |
| VAE-EBLVM [18] | **30.1** | - |
| CoopNets [61] | 33.61 | - |
| EBM (ensemble) [10] | 38.2 | - |
| MDSM$^\ddagger$ [38] | - | 31.7 |
| MDSM$^\ddagger$ (our code) | 39.12 | $30.19 \pm 2.60^\dagger$ |
| BiMDSM$^\ddagger$ (20) | **34.55** | $\mathbf{26.62} \pm 1.52^\dagger$ |
| BiMDSM$^\ddagger$ (50) | 38.82 | $29.43 \pm 2.76^\dagger$ |
| BiMDSM$^\ddagger$ (100) | 36.13 | $\mathbf{26.90} \pm 2.14^\dagger$ |

convergence is still relatively small. We also show the learning curves of the higher level problem in Fig. 5 (a-b), which deceases stably. We provide results on conditionally sampling, feature embedding and classification in Appendix C.2.

## 6 Conclusion and Discussion

We consider to extend score matching (SM) to learn energy-based latent variable models with a minimal model assumption. We reformulate a SM objective as a bi-level optimization problem, named bi-level score matching (BiSM), which introduces a variational distribution for posterior inference. We prove the equivalence of BiSM to SM under the nonparametric assumption. We develop an efficient stochastic optimization algorithm with gradient unrolling for BiSM and provide a formal convergence analysis. We show the promise of BiSM in Gaussian restricted Boltzmann machines and a highly nonstructural EBLVM parameterized by deep neural networks. BiSM is comparable to the widely adopted contrastive divergence and SM methods when they are applicable; and can learn complex EBLVMs with intractable posteriors to generate natural images.

Though promising, the bilevel optimization involved in BiSM is complicated and time-consuming. A potential future work is to simplify BiSM by directly approximating the gradient of the score function with respect to the model parameters in order to avoid such bilevel formulation.

## Broader Impact

In many real world applications, such as bioinformatics, social network analysis and so on, sometimes energy-based models are preferable than directed models. The ability of the proposed BiSM to learn general energy-based latent variable models can potentially benefit such applications and therefore benefit the society.

However, as a way to train deep generative models, this work can be abused to produce fake images, videos and news, similarly to the generative adversarial nets.

## Acknowledgments and Disclosure of Funding

We thank Ziyu Wang, Yucen Luo, Tianyu Pang and Yang Song for feedback on our work, and we thank Cheng Lu, Yuhao Zhou and Shihong Song for proofreading. This work was supported by the National Key Research and Development Program of China (Nos. 2017YFA0700904, 2020AAA0104304), NSFC Projects (Nos. 61620106010, 62076145, 62076147, U19B2034, U1811461, U19A2081), Beijing NSF Project (No. L172037), Beijing Academy of Artificial Intelligence (BAAI), THU-Bosch JCML center, Tsinghua-Huawei Joint Research Program, a grant from Tsinghua Institute for Guo Qiang, Tiangong Institute for Intelligent Computing, the JP Morgan Faculty Research Program and the NVIDIA NVAIL Program with GPU/DGX Acceleration. C. Li was supported by the Chinese postdoctoral innovative talent support program and Shuimu Tsinghua Scholar.

## Footnotes

[2] See Sec. 4 for possible extensions on discrete $\boldsymbol{v}$.

[3]https://github.com/baofff/BiSM

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
