[Supplementary Material]

# Bi-level Score Matching for Learning Energy-based Latent Variable Models: Appendix

**Fan Bao**[*]**, Chongxuan Li**[*]**, Kun Xu, Hang Su**[†]**, Jun Zhu**[†]**, Bo Zhang**
Dept. of Comp. Sci. & Tech., Institute for AI, THBI Lab, BNRist Center,
State Key Lab for Intell. Tech. & Sys., Tsinghua University, Beijing, China
bf19@mails.tsinghua.edu.cn,{chongxuanli1991, kunxu.thu}@gmail.com,
{suhangss, dcszj, dcszb}@tsinghua.edu.cn

## A  Proofs and Derivations

### A.1  Proof of Theorem 1 and Further Analysis of the Potential Bias

**Theorem 1.** *Assuming that $\forall \boldsymbol{\theta} \in \Theta$, $\exists \boldsymbol{\phi} \in \Phi$ such that $\mathcal{D}(q(\boldsymbol{h}|\boldsymbol{v};\boldsymbol{\phi})||p(\boldsymbol{h}|\boldsymbol{v};\boldsymbol{\theta})) = 0, \forall \boldsymbol{v} \in \mathrm{supp}(q)$, we have $\nabla_{\boldsymbol{\theta}}\mathcal{J}(\boldsymbol{\theta}) = \nabla_{\boldsymbol{\theta}}\mathcal{J}_{Bi}(\boldsymbol{\theta}, \boldsymbol{\phi}^*(\boldsymbol{\theta}))$.*

*Proof.* Suppose $\boldsymbol{\theta} \in \Theta$, $\boldsymbol{\phi}_0 \in \Phi$ satisfies that $q(\boldsymbol{h}|\boldsymbol{v};\boldsymbol{\phi}_0) = p(\boldsymbol{h}|\boldsymbol{v};\boldsymbol{\theta})$ for all $\boldsymbol{v} \in \mathrm{supp}(q)$, then $\mathcal{G}(\boldsymbol{\theta}, \boldsymbol{\phi}_0) = \mathbb{E}_{q(\boldsymbol{v},\epsilon)}\mathcal{D}(q(\boldsymbol{h}|\boldsymbol{v};\boldsymbol{\phi}_0)||p(\boldsymbol{h}|\boldsymbol{v};\boldsymbol{\theta})) = 0$. By the definition of $\boldsymbol{\phi}^*(\boldsymbol{\theta})$, we have $0 \le \mathcal{G}(\boldsymbol{\theta}, \boldsymbol{\phi}^*(\boldsymbol{\theta})) \le \mathcal{G}(\boldsymbol{\theta}, \boldsymbol{\phi}_0) = 0$, and thereby $\mathcal{G}(\boldsymbol{\theta}, \boldsymbol{\phi}^*(\boldsymbol{\theta})) = 0$. It means that $\boldsymbol{\phi}^*(\boldsymbol{\theta})$ also satisfies that $q(\boldsymbol{h}|\boldsymbol{v};\boldsymbol{\phi}^*(\boldsymbol{\theta})) = p(\boldsymbol{h}|\boldsymbol{v},\boldsymbol{\theta})$ for all $\boldsymbol{v} \in \mathrm{supp}(q)$. Finally, we have

$$
\begin{aligned}
\mathcal{J}_{Bi}(\boldsymbol{\theta}, \boldsymbol{\phi}^*(\boldsymbol{\theta})) =& \mathbb{E}_{q(\boldsymbol{v},\epsilon)}\mathbb{E}_{q(\boldsymbol{h}|\boldsymbol{v};\boldsymbol{\phi})}\mathcal{F}\left(\nabla_{\boldsymbol{v}}\log\frac{\tilde{p}(\boldsymbol{v},\boldsymbol{h};\boldsymbol{\theta})}{q(\boldsymbol{h}|\boldsymbol{v};\boldsymbol{\phi})}, \epsilon, \boldsymbol{v}\right)\Big|_{\boldsymbol{\phi}=\boldsymbol{\phi}^*(\boldsymbol{\theta})} \\
=& \mathbb{E}_{q(\boldsymbol{v},\epsilon)}\mathbb{E}_{p(\boldsymbol{h}|\boldsymbol{v};\boldsymbol{\theta})}\mathcal{F}\left(\nabla_{\boldsymbol{v}}\log\frac{\tilde{p}(\boldsymbol{v},\boldsymbol{h};\boldsymbol{\theta})}{p(\boldsymbol{h}|\boldsymbol{v};\boldsymbol{\theta})}, \epsilon, \boldsymbol{v}\right) \\
=& \mathbb{E}_{q(\boldsymbol{v},\epsilon)}\mathbb{E}_{q(\boldsymbol{h}|\boldsymbol{v};\boldsymbol{\phi})}\mathcal{F}\left(\nabla_{\boldsymbol{v}}\log\tilde{p}(\boldsymbol{v};\boldsymbol{\theta}), \epsilon, \boldsymbol{v}\right) \\
=& \mathbb{E}_{q(\boldsymbol{v},\epsilon)}\mathcal{F}\left(\nabla_{\boldsymbol{v}}\log\tilde{p}(\boldsymbol{v};\boldsymbol{\theta}), \epsilon, \boldsymbol{v}\right) = \mathcal{J}(\boldsymbol{\theta}),
\end{aligned}
$$

and thereby $\nabla_{\boldsymbol{\theta}}\mathcal{J}(\boldsymbol{\theta}) = \nabla_{\boldsymbol{\theta}}\mathcal{J}_{Bi}(\boldsymbol{\theta}, \boldsymbol{\phi}^*(\boldsymbol{\theta}))$. $\square$

When the assumptions in Theorem 1 don't hold, we can still bound the bias between $\mathcal{J}(\boldsymbol{\theta})$ and $\mathcal{J}_{Bi}(\boldsymbol{\theta}, \boldsymbol{\phi}^*(\boldsymbol{\theta}))$ by the minimum of $\mathcal{G}(\boldsymbol{\theta}, \boldsymbol{\phi})$ up to a constant under the following surrogate assumptions:

1. There exists a set of conditional densities $R = \{r(\boldsymbol{h}|\boldsymbol{v};\boldsymbol{\eta}) : \boldsymbol{\eta} \in H\}$ parameterized by $\boldsymbol{\eta}$ including both $\{p(\boldsymbol{h}|\boldsymbol{v};\boldsymbol{\theta})|\boldsymbol{\theta} \in \Theta\}$ and $\{q(\boldsymbol{h}|\boldsymbol{v};\boldsymbol{\phi})|\boldsymbol{\phi} \in \Phi\}$, and the divergence between two conditional densities in $R$ can be bounded by the distance of their parameterizations from below, i.e., $\exists C_1 > 0, \forall \boldsymbol{\eta}_1, \boldsymbol{\eta}_2 \in H, C_1||\boldsymbol{\eta}_1 - \boldsymbol{\eta}_2|| \le \mathbb{E}_{q(\boldsymbol{v},\epsilon)}\mathcal{D}(r(\boldsymbol{h}|\boldsymbol{v};\boldsymbol{\eta}_1)||r(\boldsymbol{h}|\boldsymbol{v};\boldsymbol{\eta}_2))$.

2. $\mathcal{J}'_{Bi}(\boldsymbol{\theta}, \boldsymbol{\eta}) := \mathbb{E}_{q(\boldsymbol{v},\epsilon)}\mathbb{E}_{r(\boldsymbol{h}|\boldsymbol{v};\boldsymbol{\eta})}\mathcal{F}\left(\nabla_{\boldsymbol{v}}\log\frac{\tilde{p}(\boldsymbol{v},\boldsymbol{h};\boldsymbol{\theta})}{r(\boldsymbol{h}|\boldsymbol{v};\boldsymbol{\eta})}, \epsilon, \boldsymbol{v}\right)$ is Lipschitz continuous w.r.t. $\boldsymbol{\eta}$ on $H$, with $C_2$ as its Lipschitz constant, and $C_2$ is independent of $\boldsymbol{\theta}$.

---

[*]Equal contribution. [†] Corresponding author.

Based on assumption 1, there exists a mapping $T_p$ from $\Theta$ to $H$ and a mapping $T_q$ from $\Phi$ to $H$, s.t. $p(\boldsymbol{h}|\boldsymbol{v};\boldsymbol{\theta}) = r(\boldsymbol{h}|\boldsymbol{v};T_p(\boldsymbol{\theta}))$ and $q(\boldsymbol{h}|\boldsymbol{v};\boldsymbol{\phi}) = r(\boldsymbol{h}|\boldsymbol{v};T_q(\boldsymbol{\phi}))$. The bias can be bounded as

$$
\begin{aligned}
|\mathcal{J}_{Bi}(\boldsymbol{\theta},\boldsymbol{\phi}^*(\boldsymbol{\theta})) - \mathcal{J}(\boldsymbol{\theta})| =& |\mathcal{J}'_{Bi}(\boldsymbol{\theta},T_q(\boldsymbol{\phi}^*(\boldsymbol{\theta}))) - \mathcal{J}'_{Bi}(\boldsymbol{\theta},T_p(\boldsymbol{\theta}))| \\
\leq& C_2 ||T_q(\boldsymbol{\phi}^*(\boldsymbol{\theta})) - T_p(\boldsymbol{\theta})|| \\
\leq& \frac{C_2}{C_1}\mathbb{E}_{q(\boldsymbol{v},\boldsymbol{\epsilon})}\mathcal{D}(r(\boldsymbol{h}|\boldsymbol{v};T_q(\boldsymbol{\phi}^*(\boldsymbol{\theta})))||r(\boldsymbol{h}|\boldsymbol{v};T_p(\boldsymbol{\theta}))) \\
=& \frac{C_2}{C_1}\mathbb{E}_{q(\boldsymbol{v},\boldsymbol{\epsilon})}\mathcal{D}(q(\boldsymbol{h}|\boldsymbol{v};\boldsymbol{\phi}^*(\boldsymbol{\theta})||p(\boldsymbol{h}|\boldsymbol{v};\boldsymbol{\theta})) \\
=& \frac{C_2}{C_1}\mathcal{G}(\boldsymbol{\theta},\boldsymbol{\phi}^*(\boldsymbol{\theta})) = \frac{C_2}{C_1}\min_{\boldsymbol{\phi}\in\Phi}\mathcal{G}(\boldsymbol{\theta},\boldsymbol{\phi}).
\end{aligned}
$$

Thereby, to ensure $|\mathcal{J}_{Bi}(\boldsymbol{\theta},\boldsymbol{\phi}^*(\boldsymbol{\theta})) - \mathcal{J}(\boldsymbol{\theta})| < \delta$, it's enough to ensure $\min\limits_{\boldsymbol{\phi}\in\Phi}\mathcal{G}(\boldsymbol{\theta},\boldsymbol{\phi}) < \frac{C_1}{C_2}\delta$. We notice that the assumption does not necessarily hold, especially in the context of deep learning and we leave a deeper analysis for the future work.

## A.2 Derivation of Divergences used in the Lower Level Problem

We now derive the equivalent forms of divergences used in the lower level optimization. If the KL divergence is used, we have:

$$
\begin{aligned}
\mathcal{D}_{KL}\left(q(\boldsymbol{h}|\boldsymbol{v};\boldsymbol{\phi})||p(\boldsymbol{h}|\boldsymbol{v};\boldsymbol{\theta})\right) =& \mathbb{E}_{q(\boldsymbol{h}|\boldsymbol{v};\boldsymbol{\phi})}\log\frac{q(\boldsymbol{h}|\boldsymbol{v};\boldsymbol{\phi})}{p(\boldsymbol{h}|\boldsymbol{v};\boldsymbol{\theta})} \\
=& \mathbb{E}_{q(\boldsymbol{h}|\boldsymbol{v};\boldsymbol{\phi})}\log\frac{q(\boldsymbol{h}|\boldsymbol{v};\boldsymbol{\phi})p(\boldsymbol{v};\boldsymbol{\theta})\mathcal{Z}(\boldsymbol{\theta})}{\tilde{p}(\boldsymbol{v},\boldsymbol{h};\boldsymbol{\theta})} \\
=& \mathbb{E}_{q(\boldsymbol{h}|\boldsymbol{v};\boldsymbol{\phi})}\left[\log\frac{q(\boldsymbol{h}|\boldsymbol{v};\boldsymbol{\phi})}{\tilde{p}(\boldsymbol{v},\boldsymbol{h};\boldsymbol{\theta})}\right] + \log p(\boldsymbol{v};\boldsymbol{\theta}) + \log\mathcal{Z}(\boldsymbol{\theta}) \\
\equiv& \mathbb{E}_{q(\boldsymbol{h}|\boldsymbol{v};\boldsymbol{\phi})}\log\frac{q(\boldsymbol{h}|\boldsymbol{v};\boldsymbol{\phi})}{\tilde{p}(\boldsymbol{v},\boldsymbol{h};\boldsymbol{\theta})},
\end{aligned}
$$

where the last equivalence holds because we optimize the divergence only with respect to $\boldsymbol{\phi}$.

If the Fisher divergence is used, we have:

$$
\begin{aligned}
&\mathcal{D}_F\left(q(\boldsymbol{h}|\boldsymbol{v};\boldsymbol{\phi})||p(\boldsymbol{h}|\boldsymbol{v};\boldsymbol{\theta})\right) \\
=& \frac{1}{2}\mathbb{E}_{q(\boldsymbol{h}|\boldsymbol{v};\boldsymbol{\phi})}\left[||\nabla_{\boldsymbol{h}}\log q(\boldsymbol{h}|\boldsymbol{v};\boldsymbol{\phi}) - \nabla_{\boldsymbol{h}}\log p(\boldsymbol{v},\boldsymbol{h};\boldsymbol{\theta})||_2^2\right] \\
=& \frac{1}{2}\mathbb{E}_{q(\boldsymbol{h}|\boldsymbol{v};\boldsymbol{\phi})}\left[||\nabla_{\boldsymbol{h}}\log q(\boldsymbol{h}|\boldsymbol{v};\boldsymbol{\phi}) - \nabla_{\boldsymbol{h}}\log\tilde{p}(\boldsymbol{v},\boldsymbol{h};\boldsymbol{\theta}) - \nabla_{\boldsymbol{h}}\log\mathcal{Z}(\boldsymbol{\theta})||_2^2\right] \\
=& \frac{1}{2}\mathbb{E}_{q(\boldsymbol{h}|\boldsymbol{v};\boldsymbol{\phi})}\left[||\nabla_{\boldsymbol{h}}\log q(\boldsymbol{h}|\boldsymbol{v};\boldsymbol{\phi}) - \nabla_{\boldsymbol{h}}\log\tilde{p}(\boldsymbol{v},\boldsymbol{h};\boldsymbol{\theta})||_2^2\right].
\end{aligned}
$$

## A.3 Some Mathematical Pre-knowledge for Proof of Theorem 2

Let $\boldsymbol{x}$ be a vector in $\mathbb{C}^n$ and $||\boldsymbol{x}||$ be the 2-norm of $\boldsymbol{x}$. Let $A \in \mathbb{C}^{n\times m}$ be a matrix and $||A|| := \sup\limits_{\boldsymbol{x}\in\mathbb{C}^m\setminus\{0\}}\frac{||A\boldsymbol{x}||}{||\boldsymbol{x}||}$ be the natural norm of $A$ induced by the 2-norm. Let $||f||_{Lip} := \sup\limits_{\boldsymbol{x}_1\neq\boldsymbol{x}_2\in X}\frac{||f(\boldsymbol{x}_2)-f(\boldsymbol{x}_1)||}{||\boldsymbol{x}_2-\boldsymbol{x}_1||}$ be the Lipschitz constant of a function $f$ mapping from a normed vector space (or a subset of it) to another normed vector space, and $||f||_\infty := \sup\limits_{\boldsymbol{x}\in X}||f(\boldsymbol{x})||$ be the norm superior of a function taking values in a normed vector space.

**Lemma 1.** *Suppose $A \in \mathbb{R}^{n\times n}$ is a symmetric positive semi-definite matrix, then $||A|| = \sup\limits_{\boldsymbol{x}\in\mathbb{C}^n,||\boldsymbol{x}||=1}\langle A\boldsymbol{x},\boldsymbol{x}\rangle$. Furthermore, if $A$ is invertible, then $||A^{-1}|| = \left(\inf\limits_{\boldsymbol{x}\in\mathbb{C}^n,||\boldsymbol{x}||=1}\langle A\boldsymbol{x},\boldsymbol{x}\rangle\right)^{-1}$.*

*Proof.* By the property of Hermitian matrix, we have $||A|| = \sup\limits_{x \in \mathbb{C}^n, ||x||=1} |\langle Ax, x \rangle|$. Since $A$ is positive semi-definite, we have $\langle Ax, x \rangle \geq 0$ and $||A|| = \sup\limits_{x \in \mathbb{C}^n, ||x||=1} \langle Ax, x \rangle$.

If $A$ is invertible, then

$$||A^{-1}|| = \sup_{x \in \mathbb{C}^n, x \neq 0} \frac{||A^{-1}x||}{||x||} = \sup_{x \in \mathbb{C}^n, x \neq 0} \frac{||x||}{||Ax||} = \left( \inf_{x \in \mathbb{C}^n, x \neq 0} \frac{||Ax||}{||x||} \right)^{-1}$$
$$= \left( \inf_{x \in \mathbb{C}^n, ||x||=1} ||Ax|| \right)^{-1} = \left( \inf_{x \in \mathbb{C}^n, ||x||=1} \left( \langle A^2 x, x \rangle \right)^{\frac{1}{2}} \right)^{-1}$$
$$= (\lambda_{min}(A^2))^{-\frac{1}{2}} = (\lambda_{min}(A))^{-1} = \left( \inf_{x \in \mathbb{C}^n, ||x||=1} \langle Ax, x \rangle \right)^{-1}.$$

$\square$

**Lemma 2.** *Suppose $A \subset \mathbb{R}^{n \times n}$ is a symmetric positive semi-definite matrix and $\alpha > 0$, s.t. $\alpha ||A|| \leq 1$, then $||I - \alpha A|| \leq 1$. Furthermore, if $A$ is invertible, then $||I - \alpha A|| = 1 - \alpha ||A^{-1}||^{-1}$.*

*Proof.* By the property of Hermitian matrix, we have

$$||I - \alpha A|| = \sup_{x \in \mathbb{C}^n, ||x||=1} |\langle (I - \alpha A)x, x \rangle| = \sup_{x \in \mathbb{C}^n, ||x||=1} |1 - \langle \alpha Ax, x \rangle|.$$

Since $\alpha ||A|| = \sup\limits_{x \in \mathbb{C}^n, ||x||=1} |\langle \alpha Ax, x \rangle| \leq 1$, we have

$$\sup_{x \in \mathbb{C}^n, ||x||=1} |1 - \langle \alpha Ax, x \rangle|| = \sup_{x \in \mathbb{C}^n, ||x||=1} 1 - \langle \alpha Ax, x \rangle \leq 1.$$

As a result, $||I - \alpha A|| \leq 1$. If $A$ is invertible, by Lemma 1, we have

$$\sup_{x \in \mathbb{C}^n, ||x||=1} 1 - \langle \alpha Ax, x \rangle = 1 - \alpha \inf_{x \in \mathbb{C}^n, ||x||=1} \langle Ax, x \rangle = 1 - \alpha ||A^{-1}||^{-1}$$

$\square$

### A.4 Proof of Theorem 2

For clarity, we explicitly write $\hat{\phi}^n(\theta)$ as $\hat{\phi}^n(\theta, \phi^0)$ to emphasize the dependence on $\phi^0$, and $\hat{\phi}^n(\theta, \phi^0)$ is recursively defined as

$$\hat{\phi}^n(\theta, \phi^0) = \hat{\phi}^{n-1}(\theta, \phi^0) - \alpha \frac{\partial \hat{\mathcal{G}}(\theta, \phi)}{\partial \phi}|_{\phi = \hat{\phi}^{n-1}(\theta, \phi^0)}, \tag{1}$$

where we slightly abuse the notation for simplicity and $\phi^0$ is also denoted as $\hat{\phi}^0(\theta, \phi^0)$.

Let $\hat{\mathcal{J}}_{Bi}^n(\theta, \phi^0) := \hat{\mathcal{J}}_{Bi}(\theta, \hat{\phi}^n(\theta, \phi^0))$ be the surrogate loss, we firstly build the relationship between the surrogate loss and the accurate loss $\hat{\mathcal{J}}_{Bi}(\theta, \hat{\phi}^*(\theta))$ by the following lemma.

**Lemma 3.** $\hat{\mathcal{J}}_{Bi}(\theta, \hat{\phi}^*(\theta)) = \hat{\mathcal{J}}_{Bi}^n(\theta, \hat{\phi}^*(\theta))$ *for all $n \geq 0$.*

*Proof.* Since $\frac{\partial \hat{\mathcal{G}}(\theta, \phi)}{\partial \phi}|_{\phi = \hat{\phi}^*(\theta)} = 0$, we have $\hat{\phi}^1(\theta, \hat{\phi}^*(\theta)) = \hat{\phi}^0(\theta, \hat{\phi}^*(\theta)) = \hat{\phi}^*(\theta)$. Similarly, we have $\hat{\phi}^n(\theta, \hat{\phi}^*(\theta)) = \hat{\phi}^*(\theta)$ for all $n \geq 1$ by the mathematical induction. As a result, $\hat{\mathcal{J}}_{Bi}^n(\theta, \hat{\phi}^*(\theta)) = \hat{\mathcal{J}}_{Bi}(\theta, \hat{\phi}^n(\theta, \hat{\phi}^*(\theta))) = \hat{\mathcal{J}}_{Bi}(\theta, \hat{\phi}^*(\theta))$. $\square$

We can further bound the difference between the gradient of the surrogate loss $\frac{\partial \hat{\mathcal{J}}_{Bi}(\theta, \hat{\phi}^n(\theta, \phi^0))}{\partial \theta}$ and the gradient of the true loss $\frac{\partial \hat{\mathcal{J}}_{Bi}(\theta, \hat{\phi}^*(\theta))}{\partial \theta}$ as

$$\left\|\frac{\partial \hat{\mathcal{J}}_{Bi}(\boldsymbol{\theta}, \hat{\boldsymbol{\phi}}^n(\boldsymbol{\theta}, \boldsymbol{\phi}^0))}{\partial \boldsymbol{\theta}} - \frac{\partial \hat{\mathcal{J}}_{Bi}(\boldsymbol{\theta}, \hat{\boldsymbol{\phi}}^*(\boldsymbol{\theta}))}{\partial \boldsymbol{\theta}}\right\| = \left\|\frac{\partial \hat{\mathcal{J}}_{Bi}^n(\boldsymbol{\theta}, \boldsymbol{\phi}^0)}{\partial \boldsymbol{\theta}} - \frac{\partial \hat{\mathcal{J}}_{Bi}^n(\boldsymbol{\theta}, \hat{\boldsymbol{\phi}}^*(\boldsymbol{\theta}))}{\partial \boldsymbol{\theta}}\right\|$$

$$= \left\|\frac{\partial \hat{\mathcal{J}}_{Bi}^n(\boldsymbol{\theta}, \boldsymbol{\phi}^0)}{\partial \boldsymbol{\theta}} - \frac{\partial \hat{\mathcal{J}}_{Bi}^n(\boldsymbol{\theta}, \boldsymbol{\phi}^0)}{\partial \boldsymbol{\theta}}\Big|_{\boldsymbol{\phi}^0=\hat{\boldsymbol{\phi}}^*(\boldsymbol{\theta})} - \frac{\partial \hat{\mathcal{J}}_{Bi}^n(\boldsymbol{\theta}, \boldsymbol{\phi}^0)}{\partial \boldsymbol{\phi}^0}\Big|_{\boldsymbol{\phi}^0=\hat{\boldsymbol{\phi}}^*(\boldsymbol{\theta})} \frac{\partial \hat{\boldsymbol{\phi}}^*(\boldsymbol{\theta})}{\partial \boldsymbol{\theta}}\right\|$$

$$\leq \left\|\frac{\partial \hat{\mathcal{J}}_{Bi}^n(\boldsymbol{\theta}, \boldsymbol{\phi}^0)}{\partial \boldsymbol{\theta}} - \frac{\partial \hat{\mathcal{J}}_{Bi}^n(\boldsymbol{\theta}, \boldsymbol{\phi}^0)}{\partial \boldsymbol{\theta}}\Big|_{\boldsymbol{\phi}^0=\hat{\boldsymbol{\phi}}^*(\boldsymbol{\theta})}\right\| + \left\|\frac{\partial \hat{\mathcal{J}}_{Bi}^n(\boldsymbol{\theta}, \boldsymbol{\phi}^0)}{\partial \boldsymbol{\phi}^0}\Big|_{\boldsymbol{\phi}^0=\hat{\boldsymbol{\phi}}^*(\boldsymbol{\theta})} \frac{\partial \hat{\boldsymbol{\phi}}^*(\boldsymbol{\theta})}{\partial \boldsymbol{\theta}}\right\| \qquad (2)$$

The first term in Eqn. (2) has a bound in the form of $(A + Bn)\kappa^n\|\boldsymbol{\phi}^0 - \hat{\boldsymbol{\phi}}^*(\boldsymbol{\theta})\|$ and the second term in Eqn. (2) has a bound in the form of $C\kappa^n$, as shown in Theorem 2.

**Theorem 2.** *Suppose the following assumptions hold:*

1. *Both $\Theta$ and $\Phi$ are compact and convex,*

2. *$\hat{\mathcal{J}}_{Bi}(\boldsymbol{\theta}, \boldsymbol{\phi}) \in C^2(\Omega)$, $\hat{\mathcal{G}}(\boldsymbol{\theta}, \boldsymbol{\phi}) \in C^3(\Omega)$, where $\Omega$ is an open set including $\Theta \times \Phi$ (i.e. $\hat{\mathcal{J}}_{Bi}$ and $\hat{\mathcal{G}}$ are second and third order continuously differentiable on $\Omega$ respectively),*

3. *$\hat{\mathcal{G}}(\boldsymbol{\theta}, \boldsymbol{\phi})$ is strongly convex on $\Phi$ for all $\boldsymbol{\theta} \in \Theta$,*

4. *$\forall n \geq 0, \forall \boldsymbol{\theta} \in \Theta, \forall \boldsymbol{\phi}^0 \in \Phi, \hat{\boldsymbol{\phi}}^n(\boldsymbol{\theta}, \boldsymbol{\phi}^0) \in \Phi$ and $\hat{\boldsymbol{\phi}}^*(\boldsymbol{\theta}) \in \Phi$,*

*then when $\alpha$ is small enough, there exists $A, B, C > 0$ and $\kappa \in (0, 1)$ independent of $\boldsymbol{\theta}$ and $\boldsymbol{\phi}^0$, s.t.,*

$$\left\|\frac{\partial \hat{\mathcal{J}}_{Bi}(\boldsymbol{\theta}, \hat{\boldsymbol{\phi}}^n(\boldsymbol{\theta}, \boldsymbol{\phi}^0))}{\partial \boldsymbol{\theta}} - \frac{\partial \hat{\mathcal{J}}_{Bi}(\boldsymbol{\theta}, \hat{\boldsymbol{\phi}}^*(\boldsymbol{\theta}))}{\partial \boldsymbol{\theta}}\right\| \leq (A + Bn)\kappa^n\|\boldsymbol{\phi}^0 - \hat{\boldsymbol{\phi}}^*(\boldsymbol{\theta})\| + C\kappa^n,$$

*for all $\boldsymbol{\theta} \in \Theta$, $\boldsymbol{\phi}^0 \in \Phi$ and $n \geq 0$.*

*Proof.* By assumptions 1 and 2, when $\boldsymbol{\theta} \in \Theta$ and $\boldsymbol{\phi} \in \Phi$, the norms of $k$ order ($0 \leq k \leq 2$) partial derivatives of $\hat{\mathcal{J}}_{Bi}(\boldsymbol{\theta}, \boldsymbol{\phi})$ can be bounded by a positive constant $A_1$ and the norms of $k$ order ($0 \leq k \leq 3$) partial derivatives of $\hat{\mathcal{G}}(\boldsymbol{\theta}, \boldsymbol{\phi})$ can be bounded by a positive constant $A_2$. By assumption 2 and 3, $\frac{\partial^2 \hat{\mathcal{G}}(\boldsymbol{\theta}, \boldsymbol{\phi})}{\partial \boldsymbol{\phi}^2}$ is positive definite and thereby invertible for all $\boldsymbol{\theta} \in \Theta$ and $\boldsymbol{\phi} \in \Phi$. By assumptions 1, 2, 3 and the smoothness of matrix inverse operator, we have $A_3 := \sup_{\boldsymbol{\theta} \in \Theta} \sup_{\boldsymbol{\phi} \in \Phi} \|(\frac{\partial^2 \hat{\mathcal{G}}(\boldsymbol{\theta}, \boldsymbol{\phi})}{\partial \boldsymbol{\phi}^2})^{-1}\| < \infty$.

We choose the learning rate $\alpha$ s.t. $\alpha \leq \frac{1}{A_2}$. By Lemma 2 we have

$$\left\|I - \alpha \frac{\partial^2 \hat{\mathcal{G}}(\boldsymbol{\theta}, \boldsymbol{\phi})}{\partial \boldsymbol{\phi}^2}\right\| = 1 - \alpha\|(\frac{\partial^2 \hat{\mathcal{G}}(\boldsymbol{\theta}, \boldsymbol{\phi})}{\partial \boldsymbol{\phi}^2})^{-1}\|^{-1} \leq 1 - \alpha A_3^{-1}, \quad \forall \boldsymbol{\theta} \in \Theta, \forall \boldsymbol{\phi} \in \Phi.$$

Taking partial derivative of Eqn. (1) w.r.t. $\boldsymbol{\phi}^0$, we have

$$\frac{\partial \hat{\boldsymbol{\phi}}^n(\boldsymbol{\theta}, \boldsymbol{\phi}^0)}{\partial \boldsymbol{\phi}^0} = \frac{\partial \hat{\boldsymbol{\phi}}^{n-1}(\boldsymbol{\theta}, \boldsymbol{\phi}^0)}{\partial \boldsymbol{\phi}^0} - \alpha \frac{\partial^2 \hat{\mathcal{G}}(\boldsymbol{\theta}, \boldsymbol{\phi})}{\partial \boldsymbol{\phi}^2}\Big|_{\boldsymbol{\phi}=\hat{\boldsymbol{\phi}}^{n-1}(\boldsymbol{\theta}, \boldsymbol{\phi}^0)} \frac{\partial \hat{\boldsymbol{\phi}}^{n-1}(\boldsymbol{\theta}, \boldsymbol{\phi}^0)}{\partial \boldsymbol{\phi}^0}$$

$$= (I - \alpha \frac{\partial^2 \hat{\mathcal{G}}(\boldsymbol{\theta}, \boldsymbol{\phi})}{\partial \boldsymbol{\phi}^2}\Big|_{\boldsymbol{\phi}=\hat{\boldsymbol{\phi}}^{n-1}(\boldsymbol{\theta}, \boldsymbol{\phi}^0)}) \frac{\partial \hat{\boldsymbol{\phi}}^{n-1}(\boldsymbol{\theta}, \boldsymbol{\phi}^0)}{\partial \boldsymbol{\phi}^0},$$

$$\left\|\frac{\partial \hat{\boldsymbol{\phi}}^n(\boldsymbol{\theta}, \boldsymbol{\phi}^0)}{\partial \boldsymbol{\phi}^0}\right\| \leq \left\|I - \alpha \frac{\partial^2 \hat{\mathcal{G}}(\boldsymbol{\theta}, \boldsymbol{\phi})}{\partial \boldsymbol{\phi}^2}\Big|_{\boldsymbol{\phi}=\hat{\boldsymbol{\phi}}^{n-1}(\boldsymbol{\theta}, \boldsymbol{\phi}^0)})\right\| \left\|\frac{\partial \hat{\boldsymbol{\phi}}^{n-1}(\boldsymbol{\theta}, \boldsymbol{\phi}^0)}{\partial \boldsymbol{\phi}^0}\right\|$$

$$\leq (1 - \alpha A_3^{-1})\left\|\frac{\partial \hat{\boldsymbol{\phi}}^{n-1}(\boldsymbol{\theta}, \boldsymbol{\phi}^0)}{\partial \boldsymbol{\phi}^0}\right\|, \quad \forall \boldsymbol{\theta} \in \Theta, \forall \boldsymbol{\phi}^0 \in \Phi, \forall n \geq 1.$$

Thereby, we have

$$\left\|\frac{\partial \hat{\boldsymbol{\phi}}^n(\boldsymbol{\theta}, \boldsymbol{\phi}^0)}{\partial \boldsymbol{\phi}^0}\right\| \leq (1 - \alpha A_3^{-1})^n \left\|\frac{\partial \hat{\boldsymbol{\phi}}^0(\boldsymbol{\theta}, \boldsymbol{\phi}^0)}{\partial \boldsymbol{\phi}^0}\right\| = (1 - \alpha A_3^{-1})^n, \quad \forall \boldsymbol{\theta} \in \Theta, \forall \boldsymbol{\phi}^0 \in \Phi, \forall n \geq 0.$$

Taking partial derivative of Eqn. (1) w.r.t. $\boldsymbol{\theta}$, we have

$$
\begin{aligned}
\frac{\partial \hat{\phi}^n(\boldsymbol{\theta}, \boldsymbol{\phi}^0)}{\partial \boldsymbol{\theta}} =& \frac{\partial \hat{\phi}^{n-1}(\boldsymbol{\theta}, \boldsymbol{\phi}^0)}{\partial \boldsymbol{\theta}} - \alpha \frac{\partial^2 \hat{\mathcal{G}}(\boldsymbol{\theta}, \boldsymbol{\phi})}{\partial \boldsymbol{\theta} \partial \boldsymbol{\phi}}|_{\boldsymbol{\phi}=\hat{\phi}^{n-1}(\boldsymbol{\theta}, \boldsymbol{\phi}^0)} \\
& - \alpha \frac{\partial^2 \hat{\mathcal{G}}(\boldsymbol{\theta}, \boldsymbol{\phi})}{\partial \boldsymbol{\phi}^2}|_{\boldsymbol{\phi}=\hat{\phi}^{n-1}(\boldsymbol{\theta}, \boldsymbol{\phi}^0)} \frac{\partial \hat{\phi}^{n-1}(\boldsymbol{\theta}, \boldsymbol{\phi}^0)}{\partial \boldsymbol{\theta}} \\
=& (I - \alpha \frac{\partial^2 \hat{\mathcal{G}}(\boldsymbol{\theta}, \boldsymbol{\phi})}{\partial \boldsymbol{\phi}^2}|_{\boldsymbol{\phi}=\hat{\phi}^{n-1}(\boldsymbol{\theta}, \boldsymbol{\phi}^0)}) \frac{\partial \hat{\phi}^{n-1}(\boldsymbol{\theta}, \boldsymbol{\phi}^0)}{\partial \boldsymbol{\theta}} \\
& - \alpha \frac{\partial^2 \hat{\mathcal{G}}(\boldsymbol{\theta}, \boldsymbol{\phi})}{\partial \boldsymbol{\theta} \partial \boldsymbol{\phi}}|_{\boldsymbol{\phi}=\hat{\phi}^{n-1}(\boldsymbol{\theta}, \boldsymbol{\phi}^0)},
\end{aligned} \tag{3}
$$

$$
\begin{aligned}
||\frac{\partial \hat{\phi}^n(\boldsymbol{\theta}, \boldsymbol{\phi}^0)}{\partial \boldsymbol{\theta}}|| \leq & ||I - \alpha \frac{\partial^2 \hat{\mathcal{G}}(\boldsymbol{\theta}, \boldsymbol{\phi})}{\partial \boldsymbol{\phi}^2}|_{\boldsymbol{\phi}=\hat{\phi}^{n-1}(\boldsymbol{\theta}, \boldsymbol{\phi}^0)}|| \; ||\frac{\partial \hat{\phi}^{n-1}(\boldsymbol{\theta}, \boldsymbol{\phi}^0)}{\partial \boldsymbol{\theta}}|| \\
& + \alpha ||\frac{\partial^2 \hat{\mathcal{G}}(\boldsymbol{\theta}, \boldsymbol{\phi})}{\partial \boldsymbol{\theta} \partial \boldsymbol{\phi}}|_{\boldsymbol{\phi}=\hat{\phi}^{n-1}(\boldsymbol{\theta}, \boldsymbol{\phi}^0)}|| \\
\leq & (1 - \alpha A_3^{-1}) ||\frac{\partial \hat{\phi}^{n-1}(\boldsymbol{\theta}, \boldsymbol{\phi}^0)}{\partial \boldsymbol{\theta}}|| + \alpha A_2, \quad \forall \boldsymbol{\theta} \in \Theta, \forall \boldsymbol{\phi}^0 \in \Phi, \forall n \geq 1.
\end{aligned}
$$

Thereby, we have

$$
\begin{aligned}
||\frac{\partial \hat{\phi}^n(\boldsymbol{\theta}, \boldsymbol{\phi}^0)}{\partial \boldsymbol{\theta}}|| \leq & (1 - \alpha A_3^{-1})^n (||\frac{\partial \phi^0(\boldsymbol{\theta}, \boldsymbol{\phi})}{\partial \boldsymbol{\theta}}|| - A_3 A_2) + A_3 A_2 \\
= & (1 - (1 - \alpha A_3^{-1})^n) A_3 A_2 \leq A_3 A_2, \quad \forall \boldsymbol{\theta} \in \Theta, \forall \boldsymbol{\phi}^0 \in \Phi, \forall n \geq 0.
\end{aligned}
$$

Taking partial derivative of Eqn. (3) w.r.t. $\boldsymbol{\phi}^0$, we have

$$
\begin{aligned}
\frac{\partial^2 \hat{\phi}^n(\boldsymbol{\theta}, \boldsymbol{\phi}^0)}{\partial \boldsymbol{\phi}^0 \partial \boldsymbol{\theta}} =& (-\alpha \frac{\partial^3 \hat{\mathcal{G}}(\boldsymbol{\theta}, \boldsymbol{\phi})}{\partial \boldsymbol{\phi}^3}|_{\boldsymbol{\phi}=\hat{\phi}^{n-1}(\boldsymbol{\theta}, \boldsymbol{\phi}^0)} \frac{\partial \hat{\phi}^{n-1}(\boldsymbol{\theta}, \boldsymbol{\phi}^0)}{\partial \boldsymbol{\phi}^0}) \frac{\partial \hat{\phi}^{n-1}(\boldsymbol{\theta}, \boldsymbol{\phi}^0)}{\partial \boldsymbol{\theta}} \\
& + (I - \alpha \frac{\partial^2 \hat{\mathcal{G}}(\boldsymbol{\theta}, \boldsymbol{\phi})}{\partial \boldsymbol{\phi}^2}|_{\boldsymbol{\phi}=\hat{\phi}^{n-1}(\boldsymbol{\theta}, \boldsymbol{\phi}^0)}) \frac{\partial^2 \hat{\phi}^{n-1}(\boldsymbol{\theta}, \boldsymbol{\phi}^0)}{\partial \boldsymbol{\phi}^0 \partial \boldsymbol{\theta}} \\
& - \alpha \frac{\partial^3 \hat{\mathcal{G}}(\boldsymbol{\theta}, \boldsymbol{\phi})}{\partial \boldsymbol{\phi} \partial \boldsymbol{\theta} \partial \boldsymbol{\phi}}|_{\boldsymbol{\phi}=\hat{\phi}^{n-1}(\boldsymbol{\theta}, \boldsymbol{\phi}^0)} \frac{\partial \hat{\phi}^{n-1}(\boldsymbol{\theta}, \boldsymbol{\phi}^0)}{\partial \boldsymbol{\phi}^0},
\end{aligned}
$$

$$
\begin{aligned}
||\frac{\partial^2 \hat{\phi}^n(\boldsymbol{\theta}, \boldsymbol{\phi}^0)}{\partial \boldsymbol{\phi}^0 \partial \boldsymbol{\theta}}|| \leq & \alpha ||\frac{\partial^3 \hat{\mathcal{G}}(\boldsymbol{\theta}, \boldsymbol{\phi})}{\partial \boldsymbol{\phi}^3}|_{\boldsymbol{\phi}=\hat{\phi}^{n-1}(\boldsymbol{\theta}, \boldsymbol{\phi}^0)}|| \; ||\frac{\partial \hat{\phi}^{n-1}(\boldsymbol{\theta}, \boldsymbol{\phi}^0)}{\partial \boldsymbol{\phi}^0}|| \; ||\frac{\partial \hat{\phi}^{n-1}(\boldsymbol{\theta}, \boldsymbol{\phi}^0)}{\partial \boldsymbol{\theta}}|| \\
& + ||I - \alpha \frac{\partial^2 \hat{\mathcal{G}}(\boldsymbol{\theta}, \boldsymbol{\phi})}{\partial \boldsymbol{\phi}^2}|_{\boldsymbol{\phi}=\hat{\phi}^{n-1}(\boldsymbol{\theta}, \boldsymbol{\phi}^0)}|| \; ||\frac{\partial^2 \hat{\phi}^{n-1}(\boldsymbol{\theta}, \boldsymbol{\phi}^0)}{\partial \boldsymbol{\phi}^0 \partial \boldsymbol{\theta}}|| \\
& + \alpha ||\frac{\partial^3 \hat{\mathcal{G}}(\boldsymbol{\theta}, \boldsymbol{\phi})}{\partial \boldsymbol{\phi} \partial \boldsymbol{\theta} \partial \boldsymbol{\phi}}|_{\boldsymbol{\phi}=\hat{\phi}^{n-1}(\boldsymbol{\theta}, \boldsymbol{\phi}^0)}|| \; ||\frac{\partial \hat{\phi}^{n-1}(\boldsymbol{\theta}, \boldsymbol{\phi}^0)}{\partial \boldsymbol{\phi}^0}|| \\
\leq & \alpha A_2 (1 - \alpha A_3^{-1})^{n-1} A_2 A_3 + (1 - \alpha A_3^{-1}) ||\frac{\partial^2 \hat{\phi}^{n-1}(\boldsymbol{\theta}, \boldsymbol{\phi}^0)}{\partial \boldsymbol{\phi}^0 \partial \boldsymbol{\theta}}|| \\
& + \alpha A_2 (1 - \alpha A_3^{-1})^{n-1}, \quad \forall \boldsymbol{\theta} \in \Theta, \forall \boldsymbol{\phi}^0 \in \Phi, \forall n \geq 1.
\end{aligned}
$$

Thereby, we have

$$
\begin{aligned}
||\frac{\partial^2 \hat{\phi}^n(\boldsymbol{\theta}, \boldsymbol{\phi}^0)}{\partial \boldsymbol{\phi}^0 \partial \boldsymbol{\theta}}|| \leq & n(1 - \alpha A_3^{-1})^{n-1} \alpha A_2 (A_2 A_3 + 1) + ||\frac{\partial^2 \phi^0(\boldsymbol{\theta}, \boldsymbol{\phi})}{\partial \boldsymbol{\phi}^0 \partial \boldsymbol{\theta}}|| (1 - \alpha A_3^{-1}) \\
= & n(1 - \alpha A_3^{-1})^{n-1} \alpha A_2 (A_2 A_3 + 1), \quad \forall \boldsymbol{\theta} \in \Theta, \forall \boldsymbol{\phi}^0 \in \Phi, \forall n \geq 0.
\end{aligned}
$$

The derivative of $\hat{\mathcal{J}}_{Bi}^n(\boldsymbol{\theta}, \phi^0)$ w.r.t. $\boldsymbol{\theta}$ is

$$\frac{\partial \hat{\mathcal{J}}_{Bi}^n(\boldsymbol{\theta}, \phi^0)}{\partial \boldsymbol{\theta}} = \frac{\partial \hat{\mathcal{J}}_{Bi}(\boldsymbol{\theta}, \phi)}{\partial \boldsymbol{\theta}}\big|_{\phi=\hat{\phi}^n(\boldsymbol{\theta},\phi^0)} + \frac{\partial \hat{\mathcal{J}}_{Bi}(\boldsymbol{\theta}, \phi)}{\partial \phi}\big|_{\phi=\hat{\phi}^n(\boldsymbol{\theta},\phi^0)} \frac{\partial \hat{\phi}^n(\boldsymbol{\theta}, \phi^0)}{\partial \boldsymbol{\theta}}.$$

Taking Lipschitz constant to both sides w.r.t. $\phi^0$ on $\Phi$ and by the convexity of $\Phi$, we have

$$
\begin{aligned}
||\frac{\partial \hat{\mathcal{J}}_{Bi}^n(\boldsymbol{\theta}, \cdot)}{\partial \boldsymbol{\theta}}||_{Lip} \leq & ||\frac{\partial \hat{\mathcal{J}}_{Bi}(\boldsymbol{\theta}, \cdot)}{\partial \boldsymbol{\theta}}||_{Lip}||\hat{\phi}^n(\boldsymbol{\theta}, \cdot)||_{Lip} + \sup_{\phi \in \Phi} ||\frac{\partial \hat{\mathcal{J}}_{Bi}(\boldsymbol{\theta}, \phi)}{\partial \phi}|| \, ||\frac{\partial \hat{\phi}^n(\boldsymbol{\theta}, \cdot)}{\partial \boldsymbol{\theta}}||_{Lip} \\
& + ||\frac{\partial \hat{\mathcal{J}}_{Bi}(\boldsymbol{\theta}, \cdot)}{\partial \phi}||_{Lip} \, ||\hat{\phi}^n(\boldsymbol{\theta}, \cdot)||_{Lip} \sup_{\phi^0 \in \Phi} ||\frac{\partial \hat{\phi}^n(\boldsymbol{\theta}, \phi^0)}{\partial \boldsymbol{\theta}}|| \\
\leq & \sup_{\phi \in \Phi} ||\frac{\partial^2 \hat{\mathcal{J}}_{Bi}(\boldsymbol{\theta}, \phi)}{\partial \phi \partial \boldsymbol{\theta}}|| \sup_{\phi^0 \in \Phi} ||\frac{\partial \hat{\phi}^n(\boldsymbol{\theta}, \phi^0)}{\partial \phi^0}|| \\
& + \sup_{\phi \in \Phi} ||\frac{\partial \hat{\mathcal{J}}_{Bi}(\boldsymbol{\theta}, \phi)}{\partial \phi}|| \sup_{\phi^0 \in \Phi} ||\frac{\partial^2 \hat{\phi}^n(\boldsymbol{\theta}, \phi^0)}{\partial \phi^0 \partial \boldsymbol{\theta}}|| \\
& + \sup_{\phi \in \Phi} ||\frac{\partial^2 \hat{\mathcal{J}}_{Bi}(\boldsymbol{\theta}, \phi)}{\partial \phi^2}|| \sup_{\phi^0 \in \Phi} ||\frac{\partial \hat{\phi}^n(\boldsymbol{\theta}, \phi^0)}{\partial \phi^0}|| \sup_{\phi^0 \in \Phi} ||\frac{\partial \hat{\phi}^n(\boldsymbol{\theta}, \phi^0)}{\partial \boldsymbol{\theta}}|| \\
\leq & A_1(1 - \alpha A_3^{-1})^n + A_1 n(1 - \alpha A_3^{-1})^{n-1} \alpha A_2 (A_2 A_3 + 1) \\
& + A_1(1 - \alpha A_3^{-1})^n A_3 A_2, \quad \forall \boldsymbol{\theta} \in \Theta, \forall n \geq 0.
\end{aligned}
\tag{4}
$$

As a result, we can bound the first term of Eqn. (2) as

$$
\begin{aligned}
& ||\frac{\partial \hat{\mathcal{J}}_{Bi}^n(\boldsymbol{\theta}, \phi^0)}{\partial \boldsymbol{\theta}} - \frac{\partial \hat{\mathcal{J}}_{Bi}^n(\boldsymbol{\theta}, \phi^0)}{\partial \boldsymbol{\theta}}\big|_{\phi^0=\hat{\phi}^*(\boldsymbol{\theta})}|| \\
\leq & A_1(1 + A_2 A_3)(1 + \frac{\alpha A_2}{1 - \alpha A_3^{-1}} n)(1 - \alpha A_3^{-1})^n ||\phi^0 - \hat{\phi}^*(\boldsymbol{\theta})||, \quad \forall \boldsymbol{\theta} \in \Theta, \forall \phi^0 \in \Phi, \forall n \geq 0.
\end{aligned}
\tag{5}
$$

For the second term of Eqn. (2), the partial derivative $\frac{\partial \hat{\mathcal{J}}_{Bi}^n(\boldsymbol{\theta}, \phi^0)}{\partial \phi^0}\big|_{\phi^0=\hat{\phi}^*(\boldsymbol{\theta})}$ can be expanded as

$$\frac{\partial \hat{\mathcal{J}}_{Bi}^n(\boldsymbol{\theta}, \phi^0)}{\partial \phi^0}\big|_{\phi^0=\hat{\phi}^*(\boldsymbol{\theta})} = \frac{\partial \hat{\mathcal{J}}_{Bi}(\boldsymbol{\theta}, \phi)}{\partial \phi}\big|_{\phi=\hat{\phi}^*(\boldsymbol{\theta})} \frac{\partial \hat{\phi}^n(\boldsymbol{\theta}, \phi^0)}{\partial \phi^0}\big|_{\phi^0=\phi^*(\boldsymbol{\theta})},$$

and thereby

$$
\begin{aligned}
||\frac{\partial \hat{\mathcal{J}}_{Bi}^n(\boldsymbol{\theta}, \phi^0)}{\partial \phi^0}\big|_{\phi^0=\hat{\phi}^*(\boldsymbol{\theta})}|| \leq & ||\frac{\partial \hat{\mathcal{J}}_{Bi}(\boldsymbol{\theta}, \phi)}{\partial \phi}\big|_{\phi=\hat{\phi}^*(\boldsymbol{\theta})}|| \, ||\frac{\partial \hat{\phi}^n(\boldsymbol{\theta}, \phi^0)}{\partial \phi^0}\big|_{\phi^0=\phi^*(\boldsymbol{\theta})}|| \\
\leq & A_1(1 - \alpha A_3^{-1})^n, \quad \forall \boldsymbol{\theta} \in \Theta, \forall n \geq 0.
\end{aligned}
$$

For calculating $\frac{\partial \hat{\phi}^*(\boldsymbol{\theta})}{\partial \boldsymbol{\theta}}$, we take partial derivative to $\frac{\partial \hat{\mathcal{G}}(\boldsymbol{\theta}, \phi)}{\partial \phi}\big|_{\phi=\hat{\phi}^*(\boldsymbol{\theta})} = 0$ w.r.t. $\boldsymbol{\theta}$ and get

$$\frac{\partial^2 \hat{\mathcal{G}}(\boldsymbol{\theta}, \phi)}{\partial \boldsymbol{\theta} \partial \phi}\big|_{\phi=\hat{\phi}^*(\boldsymbol{\theta})} + \frac{\partial^2 \hat{\mathcal{G}}(\boldsymbol{\theta}, \phi)}{\partial \phi^2}\big|_{\phi=\hat{\phi}^*(\boldsymbol{\theta})} \frac{\partial \hat{\phi}^*(\boldsymbol{\theta})}{\partial \boldsymbol{\theta}} = 0,$$

and thereby

$$\frac{\partial \hat{\phi}^*(\boldsymbol{\theta})}{\partial \boldsymbol{\theta}} = -(\frac{\partial^2 \hat{\mathcal{G}}(\boldsymbol{\theta}, \phi)}{\partial \phi^2}\big|_{\phi=\hat{\phi}^*(\boldsymbol{\theta})})^{-1} \frac{\partial^2 \hat{\mathcal{G}}(\boldsymbol{\theta}, \phi)}{\partial \boldsymbol{\theta} \partial \phi}\big|_{\phi=\hat{\phi}^*(\boldsymbol{\theta})},$$

$$||\frac{\partial \hat{\phi}^*(\boldsymbol{\theta})}{\partial \boldsymbol{\theta}}|| \leq ||(\frac{\partial^2 \hat{\mathcal{G}}(\boldsymbol{\theta}, \phi)}{\partial \phi^2}\big|_{\phi=\hat{\phi}^*(\boldsymbol{\theta})})^{-1}|| \, ||\frac{\partial^2 \hat{\mathcal{G}}(\boldsymbol{\theta}, \phi)}{\partial \boldsymbol{\theta} \partial \phi}\big|_{\phi=\hat{\phi}^*(\boldsymbol{\theta})}|| \leq A_3 A_2, \quad \forall \boldsymbol{\theta} \in \Theta.$$

Thus, the second term of Eqn. (2) can be bounded as

$$||\frac{\partial \hat{\mathcal{J}}_{Bi}^n(\boldsymbol{\theta}, \phi^0)}{\partial \phi^0}|_{\phi^0 = \hat{\phi}^*(\boldsymbol{\theta})} \frac{\partial \hat{\phi}^*(\boldsymbol{\theta})}{\partial \boldsymbol{\theta}}|| \leq ||\frac{\partial \hat{\mathcal{J}}_{Bi}^n(\boldsymbol{\theta}, \phi^0)}{\partial \phi^0}|_{\phi^0 = \hat{\phi}^*(\boldsymbol{\theta})}|| \, ||\frac{\partial \hat{\phi}^*(\boldsymbol{\theta})}{\partial \boldsymbol{\theta}}||$$

$$\leq A_1(1 - \alpha A_3^{-1})^n A_3 A_2, \quad \forall \boldsymbol{\theta} \in \Theta, \forall n \geq 0. \quad (6)$$

By Eqn. (2,5,6), we get

$$||\frac{\partial \hat{\mathcal{J}}_{Bi}(\boldsymbol{\theta}, \hat{\phi}^n(\boldsymbol{\theta}, \phi^0))}{\partial \boldsymbol{\theta}} - \frac{\partial \hat{\mathcal{J}}_{Bi}(\boldsymbol{\theta}, \hat{\phi}^*(\boldsymbol{\theta}))}{\partial \boldsymbol{\theta}}||$$

$$\leq A_1(1 + A_2 A_3)(1 + \frac{\alpha A_2}{1 - \alpha A_3^{-1}} n)(1 - \alpha A_3^{-1})^n ||\phi^0 - \hat{\phi}^*(\boldsymbol{\theta})||$$

$$+ A_1(1 - \alpha A_3^{-1})^n A_3 A_2, \quad \forall \boldsymbol{\theta} \in \Theta, \forall \phi^0 \in \Phi, \forall n \geq 0.$$

Let $A := A_1(1 + A_2 A_3)$, $B := A_1(1 + A_2 A_3)\frac{\alpha A_2}{1 - \alpha A_3^{-1}}$, $C := A_1 A_2 A_3$, $\kappa := 1 - \alpha A_3^{-1}$, then $A, B, C > 0$ and $\kappa \in (0, 1)$ are constants independent of $\boldsymbol{\theta}$ and $\phi^0$ and

$$||\frac{\partial \hat{\mathcal{J}}_{Bi}(\boldsymbol{\theta}, \hat{\phi}^n(\boldsymbol{\theta}, \phi^0))}{\partial \boldsymbol{\theta}} - \frac{\partial \hat{\mathcal{J}}_{Bi}(\boldsymbol{\theta}, \hat{\phi}^*(\boldsymbol{\theta}))}{\partial \boldsymbol{\theta}}||$$

$$\leq (A + Bn)\kappa^n ||\phi^0 - \hat{\phi}^*(\boldsymbol{\theta})|| + C\kappa^n, \quad \forall \boldsymbol{\theta} \in \Theta, \forall \phi^0 \in \Phi, \forall n \geq 0.$$

$\square$

## A.5 Proof of Corollary 3

**Corollary 3.** *(BiSM finds $\delta$-stationary points) For any accuracy level $\delta > 0$, assuming Theorem 2 holds, using a sufficiently large $N$, i.e. asymptotically $\mathcal{O}(\log \frac{1}{\delta})$, and a proper learning rate scheme $\beta$ [1], Algorithm 1 in the main text converges to a $\delta$-stationary point of BiSM, namely,*

$$||\frac{\partial \hat{\mathcal{J}}_{Bi}(\boldsymbol{\theta}, \hat{\phi}^*(\boldsymbol{\theta}))}{\partial \boldsymbol{\theta}}|| \leq \delta,$$

*and further a $\delta$-stationary point of SM if Theorem 1 also holds.*

*Proof.* For any $\delta > 0$ and $\boldsymbol{\theta} \in \Theta$, assuming that

$$||\frac{\partial \hat{\mathcal{J}}_{Bi}(\boldsymbol{\theta}, \hat{\phi}^*(\boldsymbol{\theta}))}{\partial \boldsymbol{\theta}}|| > \delta,$$

we have

$$2\langle \frac{\partial \hat{\mathcal{J}}_{Bi}(\boldsymbol{\theta}, \hat{\phi}^*(\boldsymbol{\theta}))}{\partial \boldsymbol{\theta}}, \frac{\partial \hat{\mathcal{J}}_{Bi}(\boldsymbol{\theta}, \hat{\phi}^N(\boldsymbol{\theta}))}{\partial \boldsymbol{\theta}} \rangle$$

$$= ||\frac{\partial \hat{\mathcal{J}}_{Bi}(\boldsymbol{\theta}, \hat{\phi}^*(\boldsymbol{\theta}))}{\partial \boldsymbol{\theta}}||^2 + ||\frac{\partial \hat{\mathcal{J}}_{Bi}(\boldsymbol{\theta}, \hat{\phi}^N(\boldsymbol{\theta}))}{\partial \boldsymbol{\theta}}||^2 - ||\frac{\partial \hat{\mathcal{J}}_{Bi}(\boldsymbol{\theta}, \hat{\phi}^*(\boldsymbol{\theta}))}{\partial \boldsymbol{\theta}} - \frac{\partial \hat{\mathcal{J}}_{Bi}(\boldsymbol{\theta}, \hat{\phi}^N(\boldsymbol{\theta}))}{\partial \boldsymbol{\theta}}||^2$$

$$\geq ||\frac{\partial \hat{\mathcal{J}}_{Bi}(\boldsymbol{\theta}, \hat{\phi}^*(\boldsymbol{\theta}))}{\partial \boldsymbol{\theta}}||^2 - ||\frac{\partial \hat{\mathcal{J}}_{Bi}(\boldsymbol{\theta}, \hat{\phi}^*(\boldsymbol{\theta}))}{\partial \boldsymbol{\theta}} - \frac{\partial \hat{\mathcal{J}}_{Bi}(\boldsymbol{\theta}, \hat{\phi}^N(\boldsymbol{\theta}))}{\partial \boldsymbol{\theta}}||^2$$

$$> \delta^2 - ||\frac{\partial \hat{\mathcal{J}}_{Bi}(\boldsymbol{\theta}, \hat{\phi}^*(\boldsymbol{\theta}))}{\partial \boldsymbol{\theta}} - \frac{\partial \hat{\mathcal{J}}_{Bi}(\boldsymbol{\theta}, \hat{\phi}^N(\boldsymbol{\theta}))}{\partial \boldsymbol{\theta}}||^2.$$

If Theorem 2 holds, using a sufficiently large $N$, i.e. asymptotically $\mathcal{O}(\log \frac{1}{\delta})$, we have

$$||\frac{\partial \hat{\mathcal{J}}_{Bi}(\boldsymbol{\theta}, \hat{\phi}^*(\boldsymbol{\theta}))}{\partial \boldsymbol{\theta}} - \frac{\partial \hat{\mathcal{J}}_{Bi}(\boldsymbol{\theta}, \hat{\phi}^N(\boldsymbol{\theta}))}{\partial \boldsymbol{\theta}}||^2 \leq \delta^2,$$

which implies that

$$\langle \frac{\partial \hat{\mathcal{J}}_{Bi}(\boldsymbol{\theta}, \hat{\phi}^*(\boldsymbol{\theta}))}{\partial \boldsymbol{\theta}}, \frac{\partial \hat{\mathcal{J}}_{Bi}(\boldsymbol{\theta}, \hat{\phi}^N(\boldsymbol{\theta}))}{\partial \boldsymbol{\theta}} \rangle > 0.$$

Therefore, using a proper learning rate scheme $\beta$ such that $\sum_{k=1}^{\infty} \beta_k = \infty$, $\sum_{k=1}^{\infty} \beta_k^2 < \infty$ [1], Algorithm 1, i.e. stochastic gradient descent based on $\frac{\partial \hat{\mathcal{J}}_{Bi}(\boldsymbol{\theta}, \hat{\boldsymbol{\phi}}^N(\boldsymbol{\theta}))}{\partial \boldsymbol{\theta}}$, will decrease $||\frac{\partial \hat{\mathcal{J}}_{Bi}(\boldsymbol{\theta}, \hat{\boldsymbol{\phi}}^*(\boldsymbol{\theta}))}{\partial \boldsymbol{\theta}}||$ in expectation until it converges to a $\delta$-stationary point of BiSM such that $||\frac{\partial \hat{\mathcal{J}}_{Bi}(\boldsymbol{\theta}, \hat{\boldsymbol{\phi}}^*(\boldsymbol{\theta}))}{\partial \boldsymbol{\theta}}|| \leq \delta$, according to Corollary 4.12 in Bottou et al. [1], whose regularity conditions are covered by the assumptions in Theorem 2. Further, if a $\delta$-stationary point of BiSM is also a $\delta$-stationary point of SM if Theorem 1 also holds. $\qquad \square$

## B   Experimental Settings

### B.1   GRBM

The batch size is 100 on both the checkerboard dataset and the Frey face dataset[2]. We train 100,000 iterations on the checkerboard dataset and 20,000 iterations on the Frey face dataset. The noise level [14] of DSM and BiDSM is 0.05 on the checkerboard dataset and 0.3 for on the Frey face dataset. The type of random directions [12] of SSM and BiSSM is the multivariate Rademacher distribution on both datasets. We choose $q(\boldsymbol{h}|\boldsymbol{v}; \boldsymbol{\phi})$ as a Bernoulli distribution for all BiSM methods and use the Gumbel-Softmax trick [7] for reparameterization of $q(\boldsymbol{h}|\boldsymbol{v}; \boldsymbol{\phi})$ with 0.1 as the temperature.

On both datasets, we tune the learning rate in $\{10^{-4}, 3 \times 10^{-4}, 10^{-3}, 3 \times 10^{-3}, 10^{-2}\}$ according to the visual quality of density plots and samples respectively. On the checkerboard dataset, all methods achieve similar results with the learning rates $10^{-3}$, $3 \times 10^{-3}$ and $10^{-2}$ and we choose $10^{-3}$ as the default value. On the Frey face dataset, we find that both DSM and BiDSM can work on the learning rate $10^{-4}$ and $3 \times 10^{-4}$ and we choose $2 \times 10^{-4}$ as the final learning rate. We also split a validation dataset from the Frey face dataset to choose the best model according to their corresponding loss on the validation dataset. We run 10 evaluations of the validation dataset during training.

On the Frey face dataset, we tune the noise level in $\{0.01, 0.03, 0.1, 0.3, 1\}$ for DSM and BiDSM and both methods only work on the noise level 0.3, so we choose 0.3 as the final noise level.

We run 1,000 steps Gibbs sampling to sample from GRBM on both datasets and all methods.

### B.2   Deep EBLVM

The batch size is 100 on both the MNIST, CIFAR10 and CelebA datasets. We scale the CelebA datasets to 32×32 and 64×64 and explicitly denote them as CelebA32 or CelebA64 when necessary. Following [9], we train 100,000 iterations on the MNIST dataset and 300,000 iterations on the CIFAR10 and the CelebA datasets; the noise level is geometrically distributed in the range $[0.1, 3.0]$ on the MNIST dataset and uniformly distributed in the range $[0.05, 1.2]$ on the CIFAR10 and the CelebA dataset; $\sigma_0$ (see [9]) is 0.1 on both datasets. We choose $q(\boldsymbol{h}|\boldsymbol{v}; \boldsymbol{\phi})$ as a Gaussian distribution parameterized by a 3-layer convolutional neural network for BiMDSM.

The energy function is $\mathcal{E}(\boldsymbol{v}, \boldsymbol{h}; \boldsymbol{\theta}) = g_3(g_2(g_1(\boldsymbol{v}; \boldsymbol{\theta}_1), \boldsymbol{h}); \boldsymbol{\theta}_2)$ for the deep EBLVM trained by BiMDSM and is $\mathcal{E}(\boldsymbol{v}; \boldsymbol{\theta}) = g_3(g_1(\boldsymbol{v}; \boldsymbol{\theta}_1); \boldsymbol{\theta}_2)$ for the fully visible deep EBM trained by the baseline MDSM. $g_1(\cdot)$ is a 12-layer ResNet for MNIST, a 18-layer ResNet for CIFAR10 and CelebA32 following [9], or a 24-layer ResNet for CelebA64. For the EBLVM, an extra fully connected layer is introduced in $g_1(\cdot)$ to match the dimension of $\boldsymbol{h}$. $g_2(\cdot, \cdot)$ is an additive coupling layer [3] to make the features output by $g_1(\cdot)$ and the latent variables strongly coupled. $g_3(\cdot)$ consists of a fully connected layer with an ELU activation function and use the square of 2-norm to output a scalar.

To sample from the deep EBLVM, we firstly resample data from the training dataset and inference their approximate posterior mean. We then sample from $p(\boldsymbol{v}|\boldsymbol{h})$ given $\boldsymbol{h}$ equal to the approximate posterior mean. Although it introduces bias due to the difference between $p(\boldsymbol{h}|\boldsymbol{v})$ and $q(\boldsymbol{h}|\boldsymbol{v})$, we find this sampling procedure can increase sample quality and diversity compared to directly sampling from $p(\boldsymbol{v}, \boldsymbol{h})$. Besides, this sampling procedure is not to reconstruct the training data since $p(\boldsymbol{v}|\boldsymbol{h})$ is multimodal, as shown in Fig. 5. To sample from deep EBM, we directly sample from $p(\boldsymbol{v})$. We use the annealed Langevin dynamics [9] as our sampling technique. Following [9], we choose $[1, 100]$ as the range of temperature and 0.02 as the step length for annealed Langevin dynamics on both EBLVM and EBM.

For MNIST, MDSM spends about 4 hours training a deep EBM and BiMDSM spends about 8 hours training a deep EBLVM. For CIFAR10, MDSM spends about 32 hours training a deep EBM and BiMDSM spends about 48 hours training a deep EBLVM. For CelebA32, BiMDSM spends about 48 hours training a deep EBLVM. The above experiments on deep EBLVMs and deep EBMs are conducted on 1 GeForce RTX 2080 Ti GPU. For CelebA64, BiMDSM spends about a week training a deep EBLVM on 4 GeForce RTX 1080 Ti GPUs.

## C   Additional Results

### C.1   GRBM

#### C.1.1   Sensitivity analysis on $N$

(a) DSM     (b) BiDSM ($N$=0)   (c) BiDSM ($N$=1)   (d) BiDSM ($N$=5)   (e) BiDSM ($N$=10)

Figure 1: Samples from GRBMs trained by DSM and BiDSM on different $N$ (0, 1, 5 and 10) according to the best validation performance on the Frey face dataset.

Fig. 1 shows samples from GRBMs trained by DSM and BiDSM on different $N$. The sample quality of BiDSM increases as $N$ increases, and is comparable to DSM when $N$=10. The result is consistent with the test Fisher divergence quantitative results in Tab. 1 in the full paper.

#### C.1.2   Sensitivity analysis on $K$

(a) DSM     (b) BiDSM ($K$=0)   (c) BiDSM ($K$=1)   (d) BiDSM ($K$=5)   (e) BiDSM ($K$=10)

| DSM | BiDSM ($K$=0) | BiDSM ($K$=1) | BiDSM ($K$=5) | BiDSM ($K$=10) |
|---|---|---|---|---|
| -5885.09 | -3775.31 | -5684.97 | -5780.18 | -5795.52 |

(f) Test Fisher divergence ↓ (subtracted by the same unknown constant only relevant to the data)

Figure 2: Samples from GRBMs trained by DSM and BiDSM on different $K$ (0, 1, 5 and 10) according to the best validation performance on the Frey face dataset.

Fig. 2 shows samples and test Fisher divergence from GRBMs trained by DSM and BiDSM on different $K$. The sample quality of BiDSM increases as $K$ increases and the test Fisher divergence decreases as $K$ increases. When $K = 0$, the variational posterior $q(\boldsymbol{h}|\boldsymbol{v}; \boldsymbol{\phi})$ doesn't change during training, leading to a much worse result than others.

### C.1.3 Sensitivity analysis on dimensions of $h$

(a) DSM (50)    (b) DSM (100)    (c) DSM (200)    (d) DSM (400)    (e) DSM (500)

(f) BiDSM (50)    (g) BiDSM (100)    (h) BiDSM (200)    (i) BiDSM (400)    (j) BiDSM (500)

|       | 50       | 100      | 200      | 400      | 500      |
|-------|----------|----------|----------|----------|----------|
| DSM   | -5703.89 | -5728.65 | -5798.64 | -5885.09 | -5895.96 |
| BiDSM | -5609.25 | -5670.93 | -5736.73 | -5800.17 | -5814.74 |

(k) Test Fisher divergence ↓ (subtracted by the same unknown constant only relevant to the data)

Figure 3: Samples (a-j) and test Fisher divergence (k) of GRBMs trained by DSM and BiDSM on different dimensions of $h$ (50, 100, 200, 400 and 500) according to the best validation performance on the Frey face dataset. $N$ is 10 for BiDSM.

Fig. 3 shows samples and test Fisher divergence from GRBM trained by DSM and BiDSM on different dimensions of $h$. Both the sample quality and test Fisher divergence of BiDSM are comparable to DSM on different dimensions of $h$.

### C.1.4 Time Complexity Comparison

Table 1: Time complexity comparison in GRBMs on the Frey face dataset. The time is the averaged training time of 100 iterations. All experiments are conducted on one GeForce GTX 1080 Ti GPU.

(a) Comparison on $N$

| Methods                 | Time (s) |
|-------------------------|----------|
| BiDSM ($N$=0, $K$=5)    | 4.35     |
| BiDSM ($N$=1, $K$=5)    | 5.07     |
| BiDSM ($N$=5, $K$=5)    | 8.61     |
| BiDSM ($N$=10, $K$=5)   | 13.78    |

(b) Comparison on $K$

| Methods                 | Time (s) |
|-------------------------|----------|
| BiDSM ($K$=1, $N$=5)    | 7.30     |
| BiDSM ($K$=2, $N$=5)    | 7.75     |
| BiDSM ($K$=5, $N$=5)    | 8.61     |
| BiDSM ($K$=10, $N$=5)   | 10.82    |

(c) Comparison between different methods

| Methods  | CD-5 | SSM  | DSM  | VNCE | BiDSM ($N$=0,$K$=5) | BiDSM($N$=$K$=5) |
|----------|------|------|------|------|---------------------|-------------------|
| Time (s) | 1.59 | 1.51 | 1.33 | 4.36 | 4.35                | 8.61              |

According to Algorithm 1, the time complexity and space complexity in a training iteration is $\mathcal{O}(K + N)$ and $\mathcal{O}(N)$ respectively. In Tab. 1 (a-b), we show the time complexity comparison of BiDSM on different $N$ (0, 1, 5 and 10) and $K$ (0, 1, 5 and 10). The training time is approximately

linearly correlated to both $N$ and $K$. In Tab. 1 (c), we show time complexity comparison between different methods. VNCE and our BiDSM are two methods of learning nonstructural EBLVMs, which require extra time to learn in a black-box manner compared to CD-5, SSM and DSM. While VNCE and BiDSM ($N$=0,$K$=5) have the similar time complexity, to the best of our knowledge VNCE hasn't been shown feasible to scale up to natural images, including the Frey face dataset (it doesn't hurt the time complexity comparison on this dataset). Besides, as stated in Appendix B.2, the training time of 100,000 iterations is 8h for BiMDSM in a deep EBLVM and 4h for MDSM in a deep EBM on MNIST; the training time of 300,000 iterations is 48h for BiMDSM in a deep EBLVM and 32h for MDSM in a deep EBM on CIFAR10. Thus, BiSM can learn general EBLVMs without a prohibitive cost.

## C.2 Deep EBLVM

### C.2.1 Sensitivity analysis on dimensions of $h$

(a) BiMDSM (20)  (b) BiMDSM (50)  (c) BiMDSM (100)

(d) BiMDSM (20)  (e) BiMDSM (50)  (f) BiMDSM (100)

(g) CelebA32 (20)  (h) CelebA32 (50)  (i) CelebA32 (100)

Figure 4: Samples from EBLVMs trained by BiMDSM on the MNIST, CIFAR10 and CelebA32 datasets with different dimensions of $h$ (20, 50 and 100).

Fig. 4 shows samples from EBLVMs trained by BiMDSM on the MNIST, CIFAR10 and CelebA32 datasets with different dimensions of $h$. The EBLVMs can produce meaningful samples in all

settings. Notice that across different dimensions of $h$ on one dataset, samples at the same position are sometimes similar because we initialize the same random seeds for different dimensions of $h$.

### C.2.2 Conditionally Sampling

(a) BiMDSM (20)        (b) BiMDSM (50)        (c) BiMDSM (100)

(d) BiMDSM (20)        (e) BiMDSM (50)        (f) BiMDSM (100)

Figure 5: Samples from conditional distribution $p(v|h)$ of EBLVMs trained by BiMDSM on the MNIST and CIFAR10 datasets with different dimensions of $h$ (20, 50 and 100).

Fig. 5 shows samples from conditional distribution $p(v|h)$ of EBLVMs trained by BiMDSM on the MNIST and CIFAR10 datasets with different dimensions of $h$. Each subfigure is split to four parts, and samples in the same part correspond to the same $h$, which is inferred from a training data via the approximate posterior mean. On each dataset, we use the same four training data in all settings to infer $h$. The samples from $p(v|h)$ are highly diverse, suggesting that $p(v|h)$ of an EBLVM is multimodal. Intrinsically, this is because the deep EBLVMs used here defines the conditional distribution $p(v|h)$ in a highly nonstructural way, in contrast to the hierarchical manner used in previous methods [8, 5, 6, 10]. Notice that it doesn't contradict the inference results in Sec. C.2.3, since $p(v|h) \propto p(v)p(h|v)$ and $p(v|h)$ can be dominated by $p(v)$.

### C.2.3 Inference Results

(a) BiMDSM (20)        (b) BiMDSM (50)        (c) BiMDSM (100)

Figure 6: t-SNE [13] embedding of the approximate posterior mean for the test MNIST data on different dimensions of $h$.

Table 2: Test classification accuracy (%) ↑ of the approximate posterior mean of EBLVMs trained by BiMDSM. We show results on the MNIST and CIFAR10 datasets with different dimensions of $h$ (20, 50 and 100). We use default linear SVM [4] implemented by sklearn as the classifier.

|  | BiMDSM (20) | BiMDSM (50) | BiMDSM (100) | Linear SVM on raw data |
|---|---|---|---|---|
| MNIST | 93.85 | 97.39 | 97.75 | 91.58 |
| CIFAR10 | 34.83 | 39.58 | 46.46 | 28.19 |

Fig. 6 shows the t-SNE [13] embedding of the approximate posterior mean for the test MNIST data on different dimensions of $h$. For MNIST, the embedding can be well separated on different dimensions of $h$. For CIFAR10, the intra-class distance can sometime be larger than the inner-class distance, and thereby the embedding can hardly be separated according to the class [2].

We train a linear SVM classifier[3] using the posterior mean learned by BiMDSM as features. Tab. 2 shows the test classification accuracy. On both datasets, the accuracy increases as the dimension of $h$ increases. The results of BiMDSM are better than a linear SVM classifier trained on raw data, suggesting that the features capture the underlying semantics of the images. We mention that previous EBLVMs [6, 10, 11] apply a supervised fine-tuning procedure to obtain better classification results. In contrast, we focus on a purely unsupervised learning setting here because our main goal is not to achieve the state-of-the-art classification results but to validate that the deep EBLVMs learned by BiSM can extract semantic features from natural images.

### C.2.4 Results on CelebA64

Figure 7: Samples of 64×64 resolution from an EBLVM trained by BiMDSM on CelebA64. The dimension of $h$ is 20.

Fig. 7 shows promising results on scaling to images of higher resolutions. We show samples of 64×64 resolution from an EBLVM trained by BiMDSM on CelebA64, which are of high diversity.

## Footnotes

[2]http://www.cs.nyu.edu/~roweis/data.html