[Reviews · NeurIPS 2020]

Review 1

Summary and Contributions: The paper proposed a bi-level variational framework to enable score matching methods to train general energy-based latent variable models. The original formulation is intractable, so the paper further propose a practical optimization algorithm based on gradient unrolling. Theoretical study (bias and convergence) on the practical algorithm and empirical study on MNIST and CIFAR-10 datasets are provided to validate the proposed method.

Strengths: The paper is very well-written and I enjoy reading the paper. The proposed variational extension of score matching and the gradient unrolling optimization seems reasonable and novel to me. It also conducted rigorous theoretical study to analyze the practical approximation of the gradient optimization. The experiments on CIFAR-10 and MNIST also demonstrate the effectiveness of BiSM.

Weaknesses: Here are some parts that the paper could potentially improve: - Some typos: e.g. in line 41-43, MLE should come first and SM should come second? - For theorem 2, it would be more interesting to explore the setting where G(theta, phi) is not strongly convex (i.e. a weaker assumption), although the assumption is acceptable if it is necessary for making things feasible. Also it seems there is a missing dependence of the bound on the batch size in theorem 2 and corollary 3, are you assuming infinite batch size here? Usually, SGD with biased gradient also depends on the batch size in a non-negligible way. - Furthermore, in line 173, I noticed that the paper update phi for K times on the same minibatch. Is this a special design? Why not use different batches (which seems to be less biased)? - Also in the paragraph following theorem 2, the paper claims the theorem provides insights into implementation. According to the theorem, the gradient estimation becomes less biased when N is larger. Is this consistent with your empirical observation? I didn't find ablation study on the hyper-parameter K. - Practical usefulness: I understand that the aim of the paper is not to establish a new SOTA. But still I wonder if the proposed method provides any additional practical benefits. It would be cool if the paper can demonstrate this. For example, is there any interesting results if we do Langevin sampling on both image space and latent space? Is it possible to do controllable image generation by manipulating or interpolating the latent variables? These make it different from a standard EBM. Also is it scalable to higher dimension such as CelebA 128x128? - Usually to make score matching work for images, we need to apply noise annealing on the images [1]. Is it necessary for the proposed method? [1] Generative Modeling by Estimating Gradients of the Data Distribution

Correctness: Yes.

Clarity: Yes, the paper is well written.

Relation to Prior Work: Yes, the paper provides preliminaries and discussions on previous works. Also some recent works on energy-based latent variable models should be discussed such as: - UNBIASED CONTRASTIVE DIVERGENCE ALGORITHM FOR TRAINING ENERGY-BASED LATENT VARIABLE MODELS

Reproducibility: Yes

Additional Feedback: Post-rebuttal feedback: I have read the author response and other reviews. I appreciate the authors' efforts on addressing our concerns. I think my concerns were mostly addressed and I hope the authors will work them into the next version.


Review 2

Summary and Contributions: This paper generalizes score matching method to learn energy-based model with latent variables. It adopts a variational approximation of the marginal log-likelihood of the visible units, and approximate the posterior of hidden units by another DNN. The learning is accomplished by alternative gradient descent with gradient unrolling. Experiments show that the proposed method is comparable to CD and SM based methods when the ground truth posterior is known, and it can model complicated EBLVM for natural images.

Strengths: - This paper provides sound theoretical analysis of the proposed method, including the relation to SM, the effectiveness of the learning and asymptotic behavior. - Experiments compare with various baseline methods (CD or SM based), validating the effectiveness of the proposed methods. - The proposed method is promising towards learning of EBLVM.

Weaknesses: - The proposed gradient unrolling method requires N steps to unrolling the gradient, which is slow and perhaps difficult to scale up to learning large and complicated EBLVMs. Although corollary 3 indicates that the estimation accuracy can be asymptoticly arbitrarily small, that requires N to be sufficiently large that may exceed the computing limit. - For comparison, at least one NCE-based method should be included. [1] shows that with a strong noise distribution, this line of work is possible to learn EBM on natural images. - In Table 2, it seems that higher dimension of h leads to worse result. Possible reason needs to be discussed. - Figure 4 shows that the learning can be unstable. [1] Flow Contrastive Estimation of Energy-Based Models

Correctness: The claims, method and empirical methodology look good to me.

Clarity: The paper is well written and easy to understand.

Relation to Prior Work: Yes it details prior work and relation to them.

Reproducibility: Yes

Additional Feedback: ===== after rebuttal ====== I have read the authors' reply, and they address my concerns about comparison to prior work. I decide to keep my original rate.


Review 3

Summary and Contributions: This paper presents a novel method for training energy-based latent variable models (EBLVM). This method is based on minimizing the fisher divergence between the model and the data distribution and is accomplished with an extension to score matching or denoising score matching. The fisher divergence is in general intractable to compute for latent variable models and the proposed method approximates it using a variational approximation to the distribution of hidden variables given the visible variables p(h|v). The authors prove that their proposed training method is consistent given some standard assumptions. The proposed approximation cannot be naively optimized given the dependence of the model objective on the parameters of the variational approximation. To address this the authors develop a further approximation which is based on a bi-level optimization. Within a given mini-batch, the variational posterior is optimized by a few steps of SGD to reach supposed optimality. Then this updated posterior (and all steps of the optimization) is used in computing gradients for the model objective. The authors train Gaussian-Bernoulli RBMs using their method on a toy dataset and small image dataset. They further train a deep EBLVM on MNIST and CIFAR10 and demonstrate that their models can draw samples of comparable quality to recent EBM variants with no latent variables.

Strengths: EBLVMs are a very interesting model class and the authors correctly note that they their study has been greatly limited to structured models due to a lack of scalable training methods. This approach constitutes the most scalable method to date (as far as I am aware) in training this class of models without highly structured energy functions. The bi-level optimization is fairly unintuitive but the author's detailed analysis of the method should convince skeptical readers that this is a sensible objective to optimize and should arrive at reasonable solutions. The results on image data are compelling and convincing, generating high quality samples. I would be very interested in a more detailed analysis of the benefits of the latent variable. Perhaps some conditional generation? Infer h given a test image and sample new v given h? This class of model opens up a wealth of interesting possibilities and applications that I am excited about.

Weaknesses: The authors neglect to compare to probably the 2 most related works I am aware of. The authors briefly mention variational noise contrastive estimation which can also be used to train models like those presented in this work. While this method has not yet been shown to scale to high dimensional image data it should be used as a comparison for the toy data at the very least. This work: "Variational Inference for Sparse and Undirected Models" Ingraham & Marks provides a method for parameter inference in EBLVMs. This method could also be used for comparison but at the very least should be included in the related work. The proposed method requires 2 inner loop optimizations (N x K) for each model gradient update. This will result in a slow down compared to comparable methods such as regular score matching. In the appendix there is a time comparison of various configurations of N and K but there is no comparison provided between standard score matching, CD, PCD, or DSM. This should be provided to ensure the reader that the method is not prohibitively costly. The section on deep EBLVMs lacks sufficient motivation. The ability to learn useful low-dimensional representations for high dimensional data is a unique feature of this class of model and should be discussed. I was surprised to find results on this were included in the supplement. I highly recommend placing these results into the main text as they are very interesting! The main quantitative measure for performance presented here is the fisher divergence. I find this to be somewhat weak. FID and IS are acceptable for MNIST and CIFAR but AIS/RAISE could be used to produce likelihood estimates. Especially on the GRBM models this is quite tractable. It seems the greatest weakness of the approach is the requirement of the bi-level inner-loop optimization. While I would not fault the authors for this, I feel like the trade-offs of the approach and some potential future directions should be discussed. The model architectures used should be more carefully described. The energy function is described but the inference model is not at all. This makes reproducibility not possible. Please add this.

Correctness: I feel the author's theorems are sufficient to demonstrate that the algorithm should properly train the models. The bias bound in theorem 2 is very nice but clearly depends on ||\phi^n - \phi^0||. Some empirical results demonstrating that this quantity is small for the models trained would be nice as I have no idea on the range of these values and how they will impact the bias. A small experiment demonstrating that this value is small and giving a numerical bound on the bias in these experiments would greatly strengthen the claims made here.

Clarity: The paper is relatively clearly written but some details are missing. Nowhere in the main text of the paper is their variational approximation explained. Was this a parameterized gaussian as in VAEs? I also found the notation some what confusing as the name q was used to represent the data distribution as well as the approximate posterior. I also found the notation around \phi confusing. When does \phi become \phi^0? Does the state of \phi persist across mini-batches? I believe the proposed algorithm is as follows: for K steps do sgd on \phi set \phi^0 = \phi for N steps do sgd on \phi^0 to result in \phi^N update \theta using \phi^0, \phi^1,..., \phi^N so this implies that K steps of sgd are done to \phi in each mini-batch. If this is incorrect I suggest clarifying further in your paper and algorithm boxes.

Relation to Prior Work: The authors carefully go over prior work and fit their contribution within this prior work well.

Reproducibility: No

Additional Feedback: Given that the EBLVM models do not outperform the EBMs without latent variables it would much more clearly motivate the work to have some experiments which demonstrate some of the nice properties of EBLVMs which utilize the latent variables. Please see my suggestions for edits int the Weaknesses and Claims sections for advice on how to improve this paper. ------------------------ After rebuttal and discussion ------------------------ While I still feel this method is quite complicated, I do believe the authors have addressed many of my concerns and demonstrated that my understanding of their work was indeed correct. I have decided to raise my score but keep my confidence level the same. I strongly suggest the authors try to improve the clarity of the algorithm for a latter version of this paper.


Review 4

Summary and Contributions: This paper proposes the bi-level score matching (BiSM) to learn energy-based latent variable models. Specifically, it reformulate a score matching objective as a bi-level optimization problem, which introduces a variational inference to compute posterior distribution. Experiments on Gaussian restricted Boltzmann machines and EBLVM parameterized by deep neural networks shows promising results of the proposed framework. The contributions are on those theoretical understanding of the algorithm, and the bi-level score matching concept.

Strengths: The paper provides multiple theoretical understanding of the proposed algorithm. The problem this paper tries to address is very relevant to the NeurIPS community. The bi-level score matching is somewhat novel but not that much.

Weaknesses: Two major weakness: (1) The method don't have sufficient quantitative or qualitative experimental results to verify the model. (2) The important competing baselines are missing. Only two methods are used for comparison in Table 2. Some relevant method, such as [a] that also learn latent EBM witj VAE, is missing. (3) A lot of important prior works about energy-based models with modern deep net as the energy functions are missing in the related works. For example, Xie 2016 [b] is the first paper on MLE of modern ConvNet-parametrized energy-based model. Also see [c] [d] [e]. The current references for EBM in the paper are not complete. [a] Han et al. CVPR 2020. Joint Training of Variational Auto-Encoder and Latent Energy-Based Model. [b] J Xie et al. ICML 2016. A theory of generative ConvNet. International Conference on Machine Learning. [c] J Xie et al. CVPR 2017. Synthesizing Dynamic Pattern by Spatial-Temporal Generative ConvNet. [d] J Xie et al. CVPR 2018. Learning Descriptor Networks for 3D Shape Synthesis and Analysis. [e] Wu. Sparse and Deep Generalizations of the FRAME Model --Annals of Mathematical Sciences and Applications (AMSA) 2018

Correctness: The empirical methodology seems correct to me.

Clarity: The paper is relatively clear and easy to follow. It is also well organized.

Relation to Prior Work: The paper misses a lot important related works from the computer vision community. For example, [a] latent EBM with VAE training which is related to the current paper. Pioneering works about EBM with energy functions parameterized by modern deep net are missing. See [b] [c] [d] [e] [f] . For example, [b] is the first paper on maximum likelihood learning of modern ConvNet-parametrized energy-based model. [c] is the first paper to give adversarial interpretation of maximum likelihood learning of ConvNet-EBM. [f] is the paper to jointly train EBM and generator. Due to the lack of a body of related works about EBM and latent EBM, the current paper is not in a good condition. --------------- [a] Han et al. CVPR 2020. Joint Training of Variational Auto-Encoder and Latent Energy-Based Model. [b] J Xie et al. ICML 2016. A theory of generative ConvNet. International Conference on Machine Learning. [c] J Xie et al. CVPR 2017. Synthesizing Dynamic Pattern by Spatial-Temporal Generative ConvNet. [d] J Xie et al. CVPR 2018. Learning Descriptor Networks for 3D Shape Synthesis and Analysis. [e] Wu. Sparse and Deep Generalizations of the FRAME Model --Annals of Mathematical Sciences and Applications (AMSA) 2018 [f] Cooperative Learning of Energy-Based Model and Latent Variable Model via MCMC Teaching (AAAI) 2018

Reproducibility: Yes

Additional Feedback: I suggest authors to revise their paper by (1) completing the related works of EBMs. (2) improve their performance of the method (3) compare with other latent EBM models. After rebuttal: After reading the reply from the authors, I decide to increase my score. Please follow the comments in your revision to complete the related works about the energy-based modeling, as well as adding relevant baseline comparison in the experiment. For example, I strongly suggest you to include a comparison with [b] i.e., EBM with vanilla ConvNet, (your current comparison with [10] in your paper is a EBM with fanny residual net energy), [a] VAE+latent EBM, and [f,g] CoopNets, which is cooperative learning of EBM with a generator. For your convenience, the FID score for Cifar 10 reported in [g] for CoopNets is 33.61. ============= [a] Han et al. CVPR 2020. Joint Training of Variational Auto-Encoder and Latent Energy-Based Model. [b] J Xie et al. ICML 2016. A theory of generative ConvNet. International Conference on Machine Learning. [f] Cooperative Learning of Energy-Based Model and Latent Variable Model via MCMC Teaching (AAAI) 2018 [g] Cooperative Training of Descriptor and Generator Networks. (TPAMI) 2018

[Author Response · NeurIPS 2020]

Table A: FID on CIFAR10. $^\dagger$ means averaged by 5 runs. Methods with $^\ddagger$ use comparable networks.

| Method | FID ↓ | FID-ES ↓ |
|---|---|---|
| Flow-CE [1*] | 37.30 | - |
| VAE-EBLVM[2*] | **30.1** | - |
| MDSM$^\ddagger$ [34] | - | 31.7 |
| MDSM$^\ddagger$ (our code) | 39.12 | $30.19 \pm 2.60^\dagger$ |
| BiMDSM$^\ddagger$ (20) | **34.55** | $\mathbf{26.62} \pm 1.52^\dagger$ |
| BiMDSM$^\ddagger$ (50) | 38.82 | $29.43 \pm 2.76^\dagger$ |
| BiMDSM$^\ddagger$ (100) | - | $\mathbf{26.90} \pm 2.14^\dagger$ |

Table B: Test log-likelihood (LL) results on Frey face. We use 2,000 chain samples in AIS.

| Method | LL ↑ |
|---|---|
| DSM | 129.23 |
| BiDSM ($N$=0) | 107.59 |
| BiDSM ($N$=1) | 110.65 |
| BiDSM ($N$=5) | 124.00 |
| BiDSM ($N$=10) | 125.72 |

Figure A: The gradient bias (i.e., the left hand side of Thm. 2) w.r.t. $N$ and $K$ in GRBM on Frey face.

We thank all reviewers for their valuable comments. We **update the FID results in Tab. A**, **validate Thm.2 in**
**Fig. A**, **add AIS results in Tab. B**, **add three baselines (VNCE [42], [1*] and [2*])**, and clarify other issues (e.g.,
computation time). Below, we first address the common concerns and then answer the detailed questions.

**Common concern (CC) 1 from R#2&R#4 Validate Thm. 2:** In Fig. A, the gradient bias decays (approximately)
exponentially w.r.t. $N$, as proved in Thm. 2. Besides, we find that as $K$ increases from 0 to 20, $||\phi^0 - \hat{\phi}^*||$ decreases
from 1.38 to 0.87. It leads to smaller bias (see Fig. A), which also agrees with Thm. 2. **CC2 from R#2&R#4&R#5**
**Benefits of EBLVM:** First, introducing latent variables can improve the sample quality (w.r.t. FID) in a fair comparison.
Indeed, we update Tab. 2 and obtain Tab. A, whose second column shows the FID with early stopping (ES) according
to the results on 1,000 samples. As stated in L290, a similar protocol is adopted in MDSM [34]. We also implement
MDSM in our code for a fair comparison. The reproduced MDSM is slightly better than the original paper [34] and
serves as a stronger baseline. Our result outperforms MDSM and [1*]. We mention that [2*] generate samples from a
VAE instead of an EBM and is less comparable. Second, the deep EBLVM is not suitable for conditional generation
because $p(\boldsymbol{v}|\boldsymbol{h};\boldsymbol{\theta})$ is multimodal. The results (Fig.4) and analysis are shown in Appendix C.2.2. We expect that the
model can serve as a benchmark and inspire new model design. **CC3 from R#3&R#4 Computation time:** The time
complexity per gradient estimate is $O(N+K)$. However, empirically, we don't need arbitrarily large $N$ and $K$. Indeed,
in the default setting, the training time per 100 iterations is 8.61s for BiDSM, 1.59s for CD-5, 4.36s for VNCE, 1.33s for
DSM in GRBM on Frey face. The training time of 300,000 iterations is 48h for BiMDSM and 32h for MDSM in deep
EBLVM on CIFAR10 (see L118 in Appendix B.2). Thus, BiSM can learn general EBLVMs without a prohibitive cost.

**To R#2. Typos:** We'll correct it in the final version. **Strongly convex assumption in Thm. 2:** Currently the assumption
is necessary. **Dependence on the batch size:** An infinite batch size is not necessary. Actually, the constants ($A$, $B$, $C$
and $\kappa$) and the learning rate $\alpha$ can be made independent of the batch size by applying assumptions 2 and 3 in Thm.
2 to $\mathbb{E}_{q(\boldsymbol{h}|\boldsymbol{v};\phi)}\mathcal{F}(\cdots)$ (Eqn. (8)) and $\mathcal{D}(\cdots)$ (Eqn. (9)) instead of $\hat{\mathcal{J}}_{Bi}$ and $\hat{\mathcal{G}}$. **Update $\phi$ for $K$ times on the same**
**minibatch:** It is a special design. According to Thm. 2, we should minimize $||\phi^0 - \hat{\phi}^*||$, where $\hat{\phi}^*$ is optimal on
a given minibatch. We update $\phi$ multiple times on the same minibatch to obtain $\phi^0$ that approximates $\hat{\phi}^*$. We'll
make it clearer. **Validating Thm. 2 and ablation study:** See the common concern 1. We'll add the ablation study
of $K$. **Practical usefulness:** See the common concern 2. **CelebA 128x128:** We obtain promising generation results
on CelebA 64x64 and are working on 128x128 data. We'll include the results. **Noise annealing on the images:** It is
necessary. Indeed, MDSM uses the annealed noise in its objective (Eqn. (5)). **Recent work:** Thanks. We'll discuss it.

**To R#3. Compare to [1*]:** Thanks, we compare to [1*] in Tab. A. **Higher dimension of $\boldsymbol{h}$ is worse:** We add a new
experiment with $\boldsymbol{h}$ dimension ($d_{\boldsymbol{h}}$) of 100 in Tab. A, which is comparable to $d_{\boldsymbol{h}} = 20$. The relatively worse results
of $d_{\boldsymbol{h}} = 50$ may be caused by the variance of training on different initial seeds. **Unstable learning:** The lower level
optimization can be slightly unstable because the distribution of EBLVM is moving during training. The higher level
optimization is stable. We'll plot it in the final version.

**To R#4. Compare to VNCE [42]:** We implement VNCE. On the toy data, its log likelihood is 0.303 nats, which is
worse than 0.319 nats of BiDSM. We'll add the curve of VNCE to Fig. 2 in the final version. **Missing reference:**
Thanks. We'll discuss this work in the final version. **Motivation of deep EBLVM:** See the common concern 2.
**Likelihood estimate:** See Tab. B. BiDSM gets closed to DSM as $N$ increases and $N \geq 5$ is sufficient. **Trades-offs**
**and future work:** Taking less than twice computation time of the regular SM (see common concern 3), BiSM can learn
deep EBLVMs. We'll discuss the future work in the final version. **Inference model:** Thanks. The inference model
is similar to the one used in VAE, as described in Appendix B.1 and B.2 (also see the code in the anonymous link in
Page 6). We will add more details in the main text. **Correctness:** See common concern 1. **Clarity:** The algorithm is
consistent to what you believe and we'll improve the clarity.

**To R#5. Experimental results and compare to [2*]:** We use the widely adopted FID (Tab. A) metric for evaluation
and compare to strong baselines [34][1*][2*]. Our updated results outperform baselines with comparable architectures
(see common concern 2). **Missing references:** We will include the missing references mentioned in the comments.
[1*] Flow contrastive estimation of EBMs.      [2*] Joint Training of Variational Auto-Encoder and Latent EBM.

[Meta-Review · NeurIPS 2020]

All reviewers agree this is interesting work that succefsully trains energy-based latent variable models with score matching. There were concerns around clarity of the algorithm, utility of latent variables, complexity of the bi-level optimization proess, and missing baselines, which should all be addressed (as promised in the rebuttal) in the final verison of the paper.